# Schwann cell plasticity regulates neuroblastic tumor cell differentiation via epidermal growth factor-like protein 8

Tamara Weiss [1,2,7], Sabine Taschner-Mandl [1,7✉], Lukas Janker [3,4], Andrea Bileck [3,4], Fikret Rifatbegovic [1], Florian Kromp [1], Helena Sorger [1], Maximilian O. Kauer[1], Christian Frech[1], Reinhard Windhager[5], Christopher Gerner [3,4], Peter F. Ambros[1,6] & Inge M. Ambros[1]

Adult Schwann cells (SCs) possess an inherent plastic potential. This plasticity allows SCs to acquire repair-specific functions essential for peripheral nerve regeneration. Here, we investigate whether stromal SCs in benign-behaving peripheral neuroblastic tumors adopt a similar cellular state. We profile ganglioneuromas and neuroblastomas, rich and poor in SC stroma, respectively, and peripheral nerves after injury, rich in repair SCs. Indeed, stromal SCs in ganglioneuromas and repair SCs share the expression of nerve repair-associated genes. Neuroblastoma cells, derived from aggressive tumors, respond to primary repair-related SCs and their secretome with increased neuronal differentiation and reduced proliferation. Within the pool of secreted stromal and repair SC factors, we identify EGFL8, a matricellular protein with so far undescribed function, to act as neuritogen and to rewire cellular signaling by activating kinases involved in neurogenesis. In summary, we report that human SCs undergo a similar adaptive response in two patho-physiologically distinct situations, peripheral nerve injury and tumor development.

[1] St. Anna Children's Cancer Research Institute (CCRI), Vienna, Austria. [2] Research Laboratory of the Department of Plastic and Reconstructive Surgery, Medical University of Vienna, Vienna, Austria. [3] Department of Analytical Chemistry, University of Vienna, Vienna, Austria. [4] Joint Metabolome Facility, University of Vienna & Medical University of Vienna, Vienna, Austria. [5] Department of Orthopedic Surgery, Medical University of Vienna, Vienna, Austria. [6] Department of Pediatrics, Medical University of Vienna, Vienna, Austria. [7] These authors contributed equally: Tamara Weiss, Sabine Taschner-Mandl. ✉email: sabine.taschner@ccri.at

Schwann cells (SCs) are the principal glia of the peripheral nervous system and evolve in close contact with neurons into peripheral nerve fibers. Reciprocal signaling between SCs and neurons regulates the survival, fate decisions, and differentiation of both cell types, but also influences their behavior in regenerative and pathological conditions[1–6]. Hence, understanding the molecular mechanisms underlying SC-neuron interaction is of utmost interest to develop effective treatment strategies for injuries and pathologies of the peripheral nervous system.

Despite being necessary for correct nerve development, SCs earned recognition because of their plasticity that allows differentiated SCs, further called adult SCs, to transform into a dedicated repair cell after peripheral nerve injury. The process is referred to as adaptive cellular reprogramming and includes profound transcriptional and morphological changes[7–9]. This phenotypical switch is mediated by dedifferentiation causing the regain of immature/precursor SC properties followed by re-differentiation into a repair-specific state[10]. The resulting repair SC phenotype is characterized by the re-expression of markers known to be upregulated in SCs during development, and by distinct repair functions and repair-associated ligands distinguishing repair SCs from adult SCs or developing SCs[11–13]. Those repair functions comprise the degradation of myelin debris, attraction of phagocytes, the formation of regeneration tracks for axon guidance, and the expression of cell surface proteins and trophic (neuroprotective and neuritogenic) factors promoting axon survival and re-growth[9,10,14–16]. We have recently provided a comprehensive transcriptomic and proteomic characterization of human repair SCs demonstrating that SCs isolated from excised peripheral nerves adopt the same repair-related phenotype and function in culture as in nerve tissue explants. These included the expression of master transcriptional regulators, such as JUN, as well as myelinophagy, phagocytosis, and antigen processing and presentation via MHC-II[17]. Importantly, transcriptomic signatures of primary repair-related SC cultures indicated the expression of a variety of neurotrophins and neuritogens and, thus, present an ideal in vitro model to study processes involving nerve repair and neuronal differentiation[17].

Interestingly, a prevalent stromal SC population is found in usually benign-behaving subtypes of peripheral neuroblastic tumors[18,19]. Peripheral neuroblastic tumors originate from trunk neural crest-derived sympathetic neuroblasts[20,21] and are categorized in neuroblastomas (NBs), ganglioneuroblastomas (GNBs), and ganglioneuromas (GNs) that represent a spectrum from NBs, the most aggressive form, to GNs, the most benign form, and GNBs, which exhibit various elements of both[20,22–24]. NB and GN subtypes are associated with distinct genomic alterations and strikingly different morphologies[20,22]. In general, NBs consist of un- or mostly poorly differentiated tumor cells and cancer-associated fibroblasts[25], whereas GNs are composed of differentiated, ganglionic-like tumor cells scattered within a dominant SC stroma[19,26]. The content of SC stroma was early recognized as a valuable prognostic factor as it correlates with the degree of tumor cell differentiation and a favorable outcome[19]. The ganglionic-like tumor cells also extend numerous neuritic processes that form entangled bundles surrounded by ensheathing stromal SCs[26]. This ganglion-like organoid morphology was assumed to arise from a bi-potent neoplastic neuroblastic precursor cell capable to differentiate along a neuronal and glial lineage[27]. Hence, an active role of stromal SCs in peripheral neuroblastic tumors has been neglected due to their supposed neoplastic origin.

Of note, we and others provided evidence for a non-tumor background of stromal SCs[1,28]. In a detailed immunohistochemical study, it was shown that the earliest appearance of stromal SCs is confined to the tumor blood vessels and connective tissue septa and not intermingled within the tumor as a clonal origin would imply[28]. Furthermore, we demonstrated the absence of numerical chromosomal aberrations in stromal SCs, while adjacent ganglionic-like tumor cells possessed a typical aneuploid genome[1,29,30]. These surprising findings argue against the hitherto presumed model of GNB/GN development based on a bi-potent neoplastic cell and support that the tumor cells are able to attract adult SCs from the nervous environment to the tumor.

In detaching the origin of stromal SCs in GNB/GN from a neoplastic cell, we realized how little we know about their nature. What is the cellular state of stromal SCs? How do they affect GNB/GN development? And why are they not manipulated by the tumor cells to support tumor progression but are associated with a benign tumor behavior/biology? We and others have shown that the aggressiveness of NB cell lines, derived from high-risk metastatic NBs, can be reduced upon exposure to SCs and their secreted factors[31–35]. Accordingly, a mouse study comparing intra- or extra-fascicularly grown tumor xenografts confirmed that NBs within the nervous environment were infiltrated by SCs and developed a less aggressive tumor phenotype[36]. However, a comprehensive analysis to assess the origin and functional characteristics of stromal SCs in tumors is still missing.

Based on the inherent plasticity of adult SCs and the yet unresolved nature of SC stroma, we speculate that GNB/GN development could be the result of a reactive/adaptive response of SCs to peripheral neuroblastic tumor cells similar to injured nerve cells. Thus, we here compared the cellular state of stromal SCs in GNs to repair SCs in injured nerves by transcriptome profiling of human GN and human injured nerve tissues. Moreover, we analyzed the effect of human primary repair-related SCs and their secreted factors on genetically diverse NB cells in co-culture studies and identified a promising candidate factor of therapeutic potential for aggressive NBs and peripheral nerve injuries.

## Results

**Transcriptome profiling revealed that ganglioneuromas contain stromal Schwann cells with a nerve repair-associated gene expression signature.** To assess the cellular state of stromal SCs, we performed a comprehensive transcriptomic analysis involving human tissues of SC stroma-rich GNs, SC stroma-poor NBs, and repair SC-containing injured nerves, alongside with cultures of primary human repair-related SCs and human NB cell lines (Supplementary Table 1). Immunofluorescence stainings of respective tissue sections for SC marker S100B determined a prevalent SC population of about 84% in injured nerves (Fig. 1a, Supplementary Fig. 1a), and of about 76% in GNs (Fig. 1b) as well as the almost complete absence of SCs in NBs (Fig. 1c, Supplementary Fig. 1a). Co-staining with neurofilament heavy polypeptide (NF200), an intermediate filament protein associated with mature neurons[37], marked axons in injured nerves that have mostly disintegrated after the degeneration period of 7 days (Fig. 1a). NF200 also stained ganglionic-like tumor cells with abundant neuritic processes in GNs (Fig. 1b). In line with the un- or poorly-differentiated state of tumor cells in NBs, hardly any NF200 signals were detected in NB tumor samples (Fig. 1c). Human repair-related SC cultures have been isolated according to our established protocol[38] and were positive for S100B, and showed the typical parallel alignment (Fig. 1d). Cultured NB cell lines highly expressed the neuronal ganglioside GD2 (Fig. 1e) that is characteristically found on tumor cells in NBs (Supplementary Fig. 1b) and only on some ganglionic-like tumor cells in GNs (Supplementary Fig. 1c).

Hierarchical clustering and principal component analysis of obtained RNA-seq data showed that biological samples derived

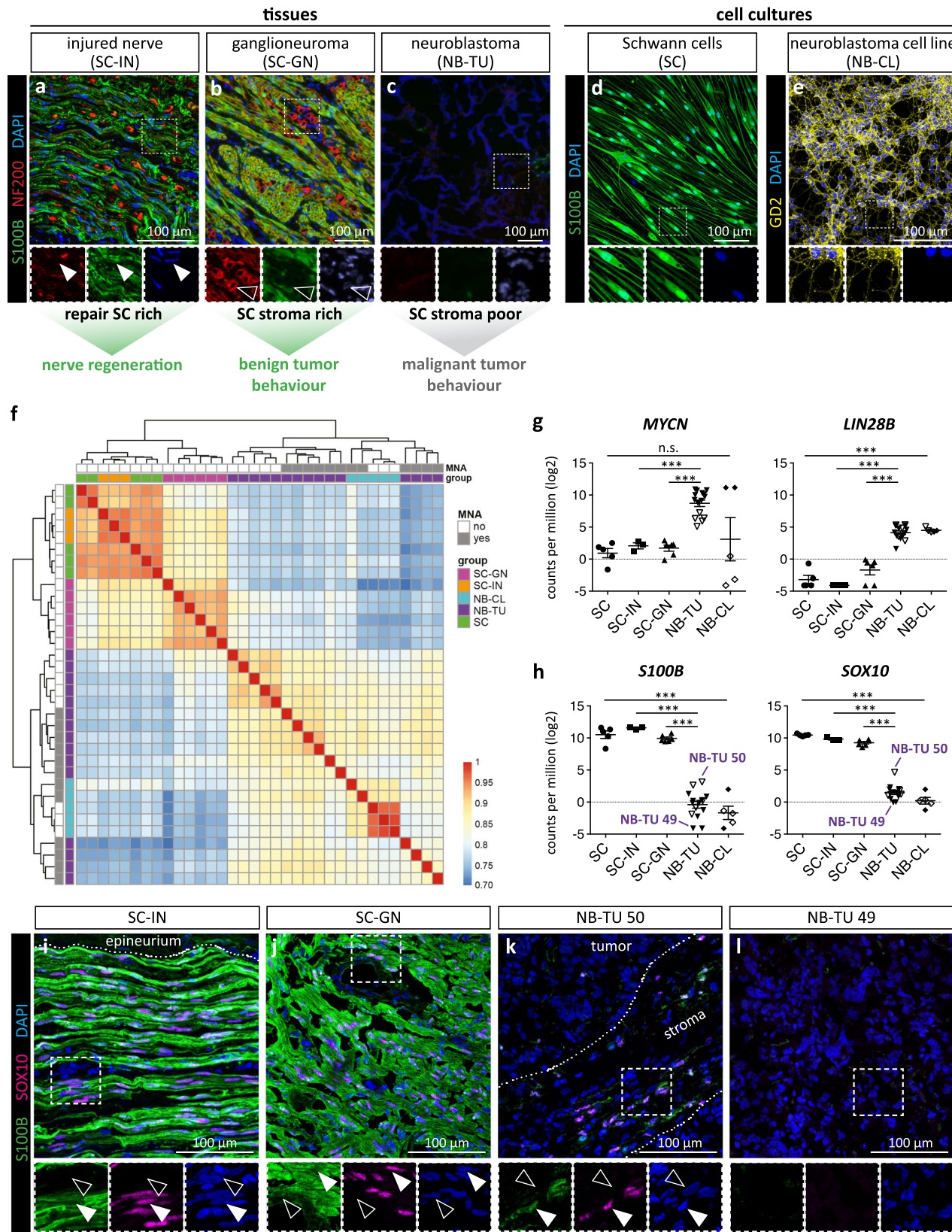

from the same tissue or cell type cluster together and that primary SCs and SC-containing tissues, i.e. injured nerves and GNs, differ from NB cell lines and NB tumors (Fig. 1f). To further confirm tissue/cell identity, we validated the expression of genes associated with either NBs, such as the miRNA suppressor *LIN28B* and the transcription factor *MYCN*[39,40], or the SC lineage, such as *S100B*

and transcription factor *SOX10*[10]. Indeed, expression of *LIN28B* was significantly higher in NBs and NB cell lines, and the *MYCN* expression level reflected the presence or absence of *MYCN* amplifications in NB cell lines and tumors (Fig. 1g, Supplementary Table 2&3). Of note, amplification of the *MYCN* oncogene is associated with an aggressive NB tumor behavior and poor

**Fig. 1 Transcriptome analysis of repair SCs in injured nerves, stromal SCs in ganglioneuromas, neuroblastomas, primary repair-related SCs, and neuroblastoma cell cultures. a–e** Tissues and cell cultures used for transcriptomic analysis. Representative immunostainings of cryosections of (**a**) injured nerve fascicle tissue (SC-IN) with S100B positive repair SCs (filled arrowheads) and NF200 positive axonal residues, (**b**) ganglioneuroma tissue (SC-GN) with S100B positive stromal SCs and NF200 positive ganglionic-like tumor cells (lined arrowheads), and (**c**) neuroblastoma tissue (NB-TU) with NF200 negative tumor cells and no SC stroma. Stainings were performed on three independent specimen per analyzed tissue. Representative immunostainings of (**d**) human primary repair-related Schwann cells (SC) positive for S100B, and (**e**) the neuroblastoma short-term cell cultures (NB-CL) CLB-Ma positive for GD2. S100B and GD2 stainings are routinely performed to characterize respective cell types. **f** RNA-seq data of SCs ($n = 5$ biological replicates from 4 donors), NB-CLs ($n = 5$ biological replicates from 3 donors), SC-GN ($n = 6$), SC-INs ($n = 3$) and NB-TU ($n = 15$) illustrated as cluster heatmap of sample-to-sample distances; computed using the Pearson correlation coefficient. Red and blue colors indicate high and low similarity between samples, respectively. Expression level of genes associated with (**g**) aggressive NBs: *MYCN* and *LIN28*, and (**h**) SCs: *S100B* and *SOX10*. Empty symbols indicate *MYCN* non-amplified, full symbols indicate *MYCN* amplified (MNA) NB-TUs and NB-CLs. *** q-value ≤ 0.001; Data are depicted as mean ± SD. Representative immunostainings of tissue cryosections of (**i**) SC-IN, (**j**) SC-GN, (**k**) NB-TU 50, and (**l**) NB-TU 49 stained for S100B, SOX10, and DAPI; filled arrowheads indicate S100B⁺/SOX10⁺ SCs, lined arrowheads indicate S100B⁻/SOX10⁻ cells. Stainings were performed on three independent specimen per analyzed tissue.

outcome[41]. The SC specific genes *S100B* and *SOX10* were significantly and strongly expressed in primary SCs, injured nerves, and GNs (Fig. 1h). Immunofluorescence stainings on tissue sections acknowledged that SOX10 positive cell nuclei correspond to S100B positive repair SCs in injured nerves (Fig. 1i, Supplementary Fig. 2a) and stromal SCs in GNs (Fig. 1j, Supplementary Fig. 2b). Moreover, the elevated level of *SOX10* mRNA found in NB-TU 50 could be ascribed to a high proportion of infiltrating stroma containing S100B and SOX10 positive SCs (Fig. 1k, Supplementary Fig. 2c), while the sections analyzed from other NB tumors such as NB-TU 49 lacked S100B and SOX10 positive cells and mRNA (Fig. 1l, Supplementary Fig. 2d).

We next defined the characteristic expression signatures of GNs and injured nerves, that both possessed a predominant SC content (Supplementary Fig. 1a), by selecting for genes significantly up-regulated (q-value > 0.05; |log2FC| >1) in GNs versus NBs, and injured nerves versus NBs. In this way, we excluded genes also present in NBs and enriched for genes characteristic for repair SCs in injured nerves and stromal SCs in GNs. Then, we compared the identified expression signatures associated with stromal SCs and repair SCs, which showed an overlap in 2755 genes (q-value > 0.05; |log2FC| >1) (Fig. 2a). Functional annotation analysis of these stromal/repair SC genes revealed pathways and gene ontology terms that could be grouped into distinct functional competences. Importantly, these functions reflected the main tasks of human repair SCs involving axon guidance, lipid/myelin degradation/metabolism, basement membrane formation/ECM (re-)organization, phagocyte attraction, and a MHC-II mediated immune regulation (Fig. 2b)[10,14,17,42]. To examine whether the expression of MHC-II is not the sole result of tissue resident immune cells, but indeed attributed to repair and stromal SCs, we stained respective tissue sections for HLA-DR and S100B. The images showed that injured nerves and GNs were highly positive for HLA-DR (Supplementary Fig. 3a,b), whereas HLA-DR staining signals were mainly restricted to the stromal portion and only scattered within the tumor cell portion of NBs (Supplementary Fig. 3c,d). Indeed, HLA-DR was expressed by S100B⁺ repair SCs in injured nerves (Supplementary Fig. 3a) as well as stromal SCs in GNs (Supplementary Fig. 3b) in addition to HLA-DR⁺/S100B⁻ immune cells (Supplementary Fig. 3a-d).

A possible repair-associated cell state of stromal SCs should be reflected by key signatures of both, developing/dedifferentiated SCs and repair-specific SCs. Accordingly, the repair/stromal SC enriched gene set included genes characteristic for SCs during development and after injury such as transcription factors *JUN*, *SOX2*, *ZEB2*, and *RUNX2* (Supplementary Fig. 4a), and receptors *NGFR*, *ERBB3*, *GFRA1*, and *CADH19*[13,43–45]. Notably, we also detected significant levels of *GDNF*, *LIF*, *SHH*, *CLCF1*, *BTC*, *CCL2*, and *UCN2* (Fig. 2c) that were reported to be exclusively expressed by repair SCs and not by adult or developing SCs[11–13] (Supplementary Fig. 4b). Moreover, *JUN* is the key transcription

factor determining the repair identity of SCs by up-regulating repair-specific target genes such as *SHH* and *GDNF*[13,46]. Hence, we performed immunofluorescence stainings for SOX10 and JUN on nerve and GN tissue sections, which confirmed that SOX10⁺ nuclei of both, repair SCs and stromal SCs were positive for JUN (Fig. 2d-f, Supplementary Fig. 5a,b). In line with the transcriptomic data, JUN was also expressed by SOX10⁻ cells such as ganglionic-like tumor cells in GNs (Fig. 2e, Supplementary Fig. 5b) and tumor cells in NBs (Fig. 2f,g, Supplementary Fig. 5c,d).

Functional annotation analysis of GN characteristic genes that were not shared with injured nerves revealed an enrichment of gene ontology terms implicated in innate immunity, inflammation as well as T- and B-cell receptor signaling pathways (Supplementary Table 5). Immunofluorescence stainings for CD3 and S100B verified the presence of CD3⁺ T-cells within the S100B⁺ SC stroma in GNs (Supplementary Fig. 6a), while CD3⁺ T-cells were only sparsely detected in the tumor cell portion of NBs (Supplementary Fig. 6b,c). In turn, genes characteristic for injured nerves not shared with GNs were assigned to gene ontology terms for the endoplasmatic reticulum, the Golgi apparatus, vesicle coating and transport, protein transport and binding, as well as acetylation and protein N-linked glycosylation (Supplementary Table 6). Those annotations suggest an active protein modification and transport machinery in repair SCs.

Taken together transcriptome profiling demonstrate that the expression signature shared by stromal SCs in GNs and repair SCs in injured nerves contain distinct nerve repair-associated genes and functions.

**Direct contact to repair-related Schwann cells promotes alignment and neurite out-growth of neuroblastoma cells.** Since we identified a repair SC-associated gene expression signature in stromal SCs, we used a co-culture model to analyze how NB cells react to repair-related SCs in vitro (Fig. 3a). Therefore, we used human primary SCs cultures as a model, as these have been shown to reflect all major characteristics of repair SCs[17]. SC cultures (passage 1) characterized by the expression of S100B, SOX10, and the intermediate filament vimentin (VIME) were used for experimentation (Fig. 3b). SCs were co-cultured with a well established human NB cell line (CLB-Ma) and short-term cultured patient-derived NB cells (STA-NB-6) alongside controls of SCs and NB cells cultured alone for 11 days. As a qualitative read-out, we established an immunofluorescence staining panel, which identified NB cells by GD2 expression and SCs by S100B expression. After 11 days, CLB-Ma and STA-NB-6 cell controls showed their typical morphology of clustered cell bodies with short, randomly extended neuritic processes (Fig. 3c,d). However, in the co-cultures with SCs, NB cells had aligned along the bi-polar SC extensions and increased the length of neuritic

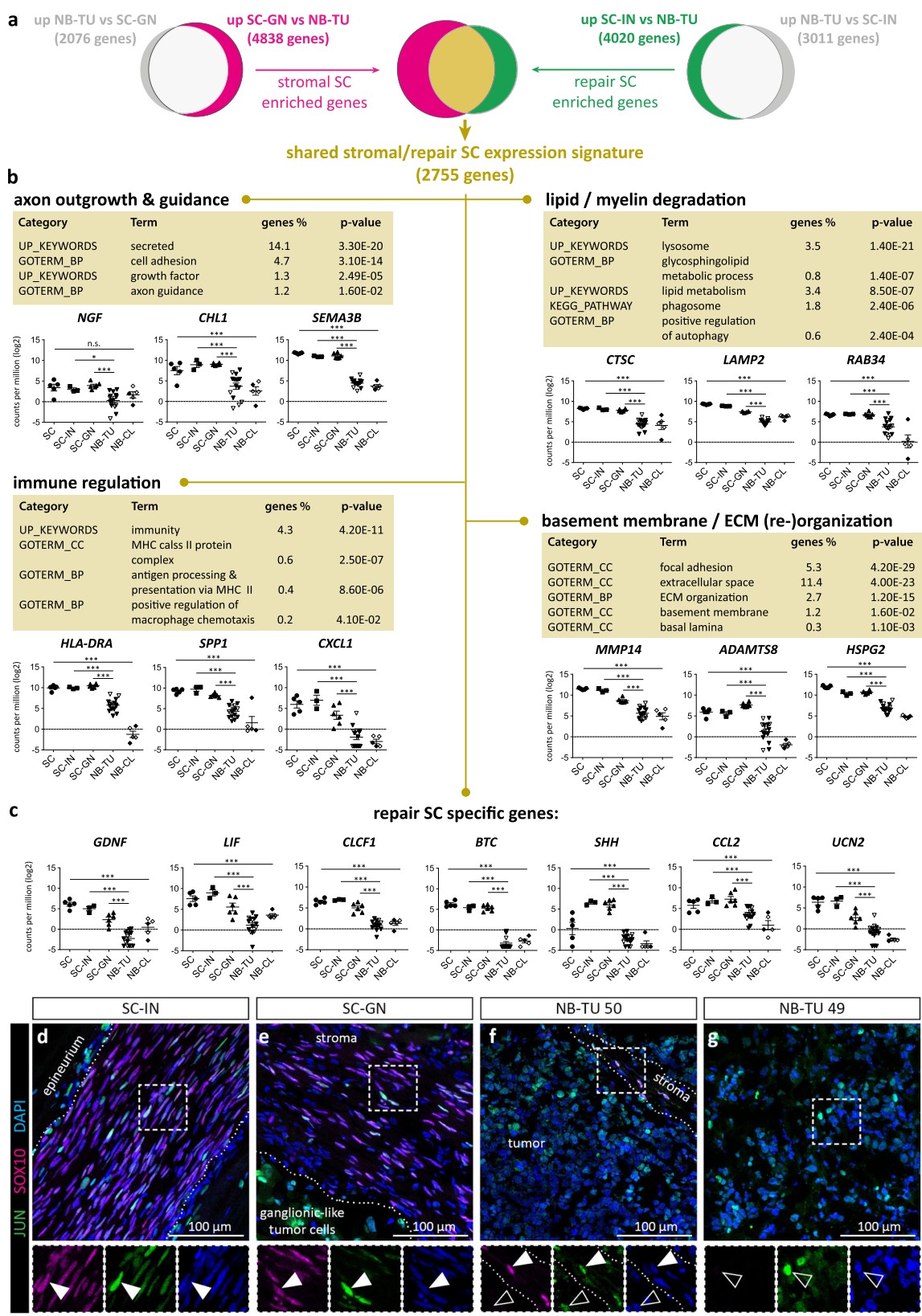

processes, predominantly in close contact with the SC surface (Fig. 3e,f,g,h arrows). Quantification of neurite length and alignment confirmed a significant increase of the mean neurite length (Fig. 3i) and neurite alignment (Fig. 3j) in co-cultures. These results suggest that the contact to human repair-related SCs induces a directed neuritic out-growth of NB cells in vitro.

**Repair-related Schwann cells induce neuronal differentiation of neuroblastoma cells independent of direct cell–cell contact.** We next aimed to dissect the effect of repair-related SCs on NB cells and distinguish signaling effects between cell bound and secreted molecules. Therefore, we refined the co-culture setting and used flow cytometry as a quantitative read-out. NB cells were either seeded in

**Fig. 2 Transcriptome profiling and functional annotation analysis of genes shared by stromal SCs in ganglioneuroma tissue and repair SCs in injured nerve tissue. a** Venn diagrams illustrate the number of significantly regulated genes (q-value > 0.05; │log2FC│ > 1) of stromal SCs (SC-GN, n = 6 independent biological replicates) and repair SCs (SC-IN, n = 3) containing tissues compared to neuroblastoma tumor tissue (NB-TU, n = 15), respectively, and the overlap in genes shared by stromal and repair SCs (q-value > 0.05; │log2FC│ >1). **b** The DAVID database [37] was used for functional annotation analysis of the 2755 gene set shared by stromal and repair SCs. KEGG pathways, functional categories (UP_KEYWORDS) and gene ontology terms (GOTERM) for biological processes (BP) and cellular compartments (CC) were manually grouped to functions such as axon outgrowth and guidance, lipid/myelin degradation, immune regulation and basement membrane/ECM (re-) organization. The expression of representative genes for each group (**b**) and the expression of specific repair SC genes (**c**) is shown for all samples; SCs (n = 5 biological replicates from 4 donors), NB-CLs (n = 5 biological replicates from 3 donors), SC-GN (n = 6), SC-INs (n = 3) and NB-TU (n = 15). Empty symbols indicate *MYCN* non-amplified NB-TUs and NB-CLs. Data are depicted as mean ± SD; *** q-value ≤ 0.001, ** q-value ≤ 0.01, * q-value ≤ 0.05, n.s. not significant. Representative immunostainings of tissue cryosections of (**d**) SC-IN, (**e**) SC-GN, (**f**) NB-TU 50, and (**g**) NB-TU 49 stained for JUN, SOX10, and DAPI; filled arrowheads indicate JUN⁺/SOX10⁺ SCs, lined arrowheads indicate JUN⁺/SOX10⁻ cells. Stainings were performed on three independent specimen per analyzed tissue.

direct contact with SCs or in a trans-well insert placed above SC cultures allowing diffusion of soluble molecules and reciprocal signaling. The refined co-culture set-up is illustrated in Fig. 4a.

In order to functionally validate whether isolated repair-related SCs reenact their key ability of regulating neuronal differentiation on NB cells in vitro, three well-established human NB cell lines (SH-SY5Y, IMR5, CLB-Ma) and two short-term NB cell cultures (STA-NB-6, STA-NB-10) covering the genetic spectrum of NBs, were co-cultured in direct and indirect contact with SCs. After 8 and 16 days, the cultures were analyzed by flow cytometry. The differentiation panel discriminated GD2⁻/S100B⁺ SCs and GD2⁺/S100B⁻ NB cells and included the neuronal differentiation marker NF200 (gating strategy Supplementary Fig. 7a). We found that NF200 expression was significantly upregulated in the *MYCN* non-amplified STA-NB-6 and SH-SY5Y NB cells after 16 days of direct contact to repair-related SCs (Fig. 4b). Of note, all NB cell lines, except STA-NB-10, showed a significant increase in NF200 expression at day 16 when co-cultured in the trans-wells without direct contact (Fig. 4b). We also noticed that the presence or absence of *MYCN* amplification in the analyzed NB cells correlated with their responsiveness to SCs (Fig. 4b). The mean fluorescence intensity histograms of NF200 further revealed that the basal NF200 expression level varied among the analyzed NB cells from low, as in CLB-Ma cells (Fig. 4c, CTRL), to highest in STA-NB-6 cells (Fig. 4d, CTRL). They also demonstrated that the increase in NF200 expression after co-culture was either due to the occurrence of a NF200⁺ subpopulation, e.g. in CLB-Ma cells (Fig. 4c, co-cultured), or an overall elevated expression, e.g. in STA-NB-6 cells (Fig. 4d, co-cultured). These findings were confirmed by qualitative assessment of NF200 expression by immunofluorescence stainings of co-cultures compared to controls of CLB-Ma cells (Fig. 4e,f) and STA-NB-6 cells (Fig. 4g,h).

To analyze whether the increase in neuronal differentiation is SC specific, we co-cultured STA-NB-6 and SH-SY5Y cells, which showed the strongest response to SCs, with immortalized human fibroblasts (iFBs) and cancer associated FBs (CAFs). After 16 days, the NF200 expression of both NB cell cultures was either unaffected or even significantly decreased upon direct and indirect contact with iFBs (Supplementary Fig. 8a) or CAFs (Supplementary Fig. 8b).

Taken together, the results demonstrate that primary repair-related SCs and/or their secreted factors are sufficient to induce neuronal differentiation of aggressive NB cell lines and primary NB cultures in vitro.

**Repair-related Schwann cells impair proliferation and increase apoptosis of neuroblastoma cells.** As cellular differentiation is accompanied by cell cycle arrest, we next determined the proliferation rate of NB cells by EdU incorporation in combination with DNA content analysis after direct and trans-well co-culture with SCs (gating strategy Supplementary Fig. 7b). Notably, after 16 days of direct SC contact the number of NB cells in the

S-phase was strongly reduced in all tested NB cell cultures (Fig. 5a). The proliferation rate of trans-well co-cultures was also significantly decreased in all NB cells, except STA-NB-10, but less pronounced as upon direct contact (Fig. 5a). The strongest anti-proliferative effects were detected in *MYCN* non-amplified STA-NB-6 and SH-SY5Y cells, as well as *MYCN* amplified IMR5 cells (Fig. 5a). Representative FACS plots illustrated the reduction of proliferation in CLB-Ma cells (Fig. 5b) and almost absent proliferation in STA-NB-6 cells (Fig. 5c) after 16 days of co-culture. This was also visualized by immunofluorescence stainings of co-cultures compared to controls of CLB-Ma cells (Fig. 5d,e) and STA-NB-6 cells (Fig. 5f,g) including the proliferation marker Ki67. In contrast, direct or indirect co-cultures with iFBs and CAFs did not influence the proliferation rate of STA-NB-6 and SHSY-5Y cells (Supplementary Fig. 7c,d).

In addition to increased differentiation and impaired proliferation, also cell death contributes to the decrease of tumor cells during GN development. Hence, we performed a terminal deoxynucleotidyl transferase dUTP nick end labeling (TUNEL) assay in combination with immunofluorescence staining for GD2 and S100B to detect apoptotic NB cells in control and co-cultures (Supplementary Fig. 9a,b). Quantitative evaluation showed that the apoptosis rate of both, *MYCN* non-amplified STA-NB-6 and *MYCN* amplified CLB-Ma cells, was increased about 10% at day 11 after direct co-culture (Supplementary Fig. 9c).

These findings show that direct and/or indirect contact to repair-related SCs decreased proliferation and elevated apoptosis of NB cells. As observed for neuronal differentiation, the *MYCN* amplification status correlated with the responsiveness of NB cells to SCs and revealed STA-NB-6 as the strongest and STA-NB-10 as the weakest SC-responsive NB cell cultures tested.

**Stromal and repair Schwann cells express EGFL8, which is able to induce neurite outgrowth and neuronal differentiation of neuroblastoma cells.** After demonstrating a pro-differentiating and anti-proliferative impact of human primary repair-related SCs on NB cells in vitro, we next aimed to identify the factors able to mediate these effects. Therefore, we interrogated the set of transcripts shared by repair SCs in injured nerve tissue and stromal SCs in GN tissue for the expression of secreted factors. Factors of interest were prioritized according to literature research and whether associated receptors, if known, were expressed by NBs. The shared secretome of repair and stromal SCs included neurotrophins such as *NGF*, *BDNF* and *GDNF* that confirmed the validity of our approach (Fig. 6a). In addition, we identified further highly expressed factors of interest such as *IGFBP6*, *FGF7* and *EGFL8* (Fig. 6a). IGFBP-6 was previously reported to inhibit the growth of SH-SY5Y cells[47] and FGF7 is involved in neuromuscular junction development[48], but both factors were not yet associated with SCs. Notably, EGFL8 was recently described by us as a potential factor involved in nerve

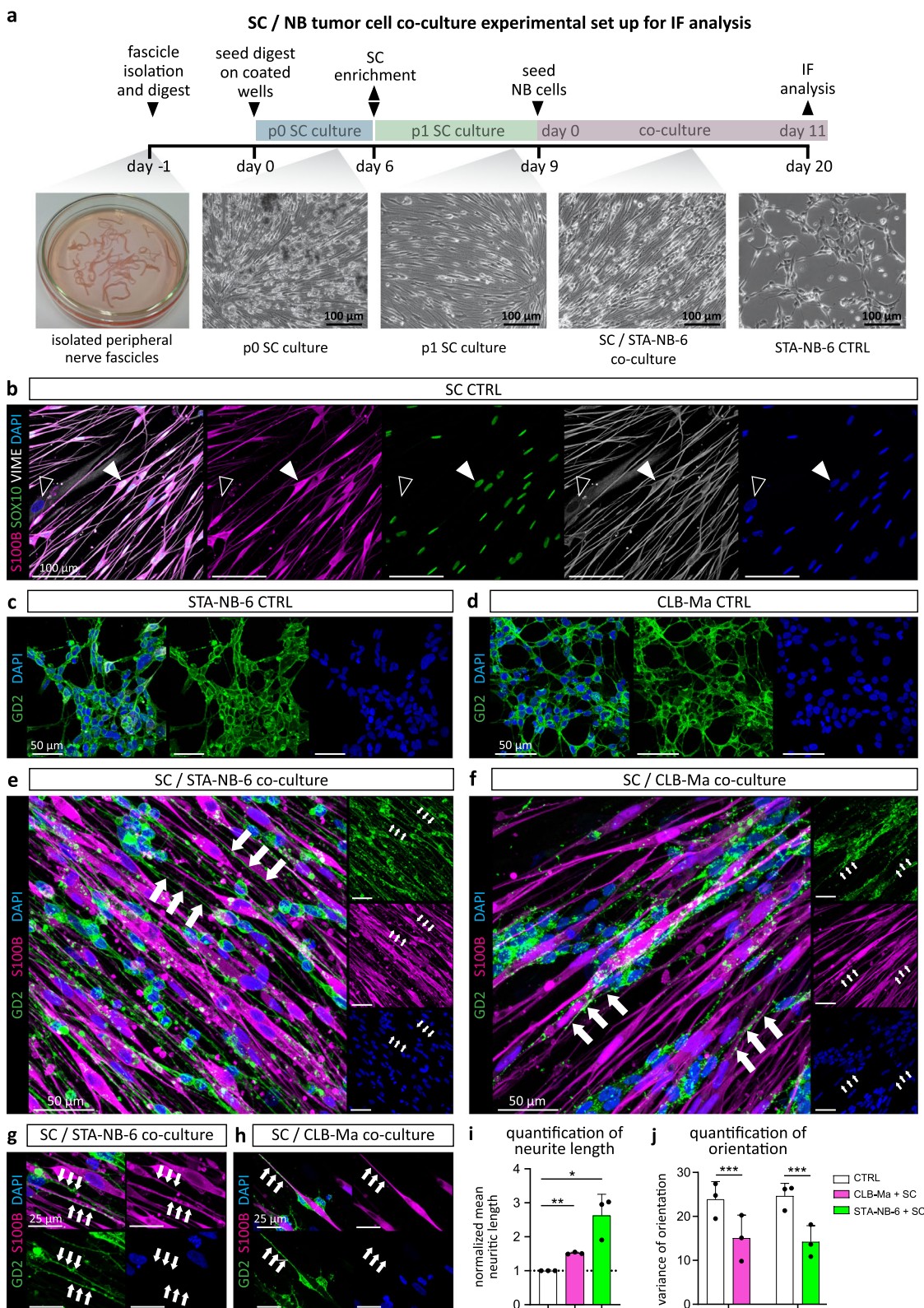

regeneration but with yet unknown function[17]. Other neurotrophic factor transcripts, such as *PTN*, highly expressed in stromal but not in repair SCs, and *CNTF*, expressed in repair but not in stromal SCs, were included in the panel of candidate factors as transcripts of their putative receptors were present in NBs (Supplementary Fig. 10).

In order to validate the effect of a set of 8 candidate factors, the recombinant proteins NGF, BDNF, GDNF, CNTF, PTN, FGF7,

IBP6 and EGFL8 were added to the SC-weakly-responsive STA-NB-10 and SC-strongly-responsive STA-NB-6 cells. Proliferation and neuronal differentiation were monitored by flow cytometry after 16 days of exposure to respective factors. As suspected, the factors had less impact on the SC-weakly-responsive STA-NB-10 cells, however, NGF and EGFL8 caused a significant anti-proliferative effect (Fig. 6b). In contrast, the SC-strongly-responsive STA-NB-6 cells were significantly impaired in

**Fig. 3 Establishment of a co-culture model to validate the effect of repair-related SCs on neuroblastoma cells in vitro. a** Scheme of SC isolation, SC culture, and SC/NB cell co-culture. Representative immunofluorescence images of (**b**) a human primary repair-related SC culture stained for S100B, SOX10, vimentin (VIME), and DAPI; filled arrowheads indicate a S100B⁺/SOX10⁺/VIME⁺ SC, lined arrowheads indicate a S100B⁻/SOX10⁻/VIME⁺ fibroblast. Stainings were performed on three independent SC cultures. Representative immunostainings of GD2⁺ STA-NB-6 (**c**) and CLB-Ma (**d**) NB cell controls as well as of (**e–h**) STA-NB-6 or CLB-Ma cells co-cultured with SCs at day 11. Arrows indicate extended neuritic processes aligned along SCs. Stainings were performed on NB cell controls and co-cultures with SCs derived from three independent donors per cell line. Quantification of (**i**) neurite length of STA-NB-6 ($p = 0.0452$) and CLB-Ma ($p = 0.002$) cells co-cultured with SCs compared to NB cell controls without SCs. Data are depicted as normalized mean neurite length ± SD ($n \geq 300$ cells over 6 images per condition over 3 independent experiments). Statistical test: repeated measures ANOVA and Dunnett's multiple comparison test. **j** Quantification of alignment of STA-NB-6 ($p < 0.0001$) and CLB-Ma ($p < 0.0001$) cells co-cultured with SCs compared to NB cell controls without SCs. Variance of orientation (variance of deviation of main cell orientation) ± SD; a value of 0 corresponds to perfect alignment; (180 datapoints in each 3 images over 3 independent experiments); For each pair of measurements (control and co-culture), a Levene test was applied to test for equal variances; *** $p$-value $\leq 0.001$, ** $p$-value $\leq 0.01$, * $p$-value $\leq 0.05$.

proliferation and showed increased neuronal differentiation after treatment with either NGF, EGFL8, BDNF, CNTF, PTN or GDNF (Fig. 6c). Notably, the effect of EGFL8 was concentration dependent and comparable to NGF, one of the most potent neurotrophins known so far (Fig. 6d). EGFL8 also acted pro-differentiating on CLB-Ma and SH-SY5Y cells, while an anti-proliferative effect was only observed in the latter (Fig. 6e). Phase contrast images illustrated the reduction of cell number and longer neuritic processes (Fig. 6f). Compared to untreated controls, STA-NB-6 cells showed a significant increase of neurite length after NGF- and EGFL8-treatment (Fig. 6g).

These findings demonstrate that EGFL8, a protein so far only described in thymocyte development[49], has a neuritogenic function able to enhance neuronal differentiation and/or to impair proliferation of aggressive NB cells.

**EGFL8 gene expression level in neuroblastomas correlates with increased patient survival.** As EGFL8 exerted anti-tumor activity on NB cells in vitro, we next assessed whether *EGFL8* expression levels in peripheral neuroblastic tumors may correlate with the clinical outcome. Analysis of the overall patient survival according to *EGFL8* gene expression was performed using the *R2: Genomics Analysis and Visualization platform*. Two different datasets, comprising 649 and 283 tumor specimens, respectively, demonstrated an over 90% and 70%, respectively, 5-year overall survival probability for patients with high *EGFL8* expression, but less than 60% and 40%, respectively, for patients with low *EGFL8* expression (Fig. 6h & Supplementary Fig. 11 a,b). Information about the stromal SC content of the included tumor specimens was not available. These data show that *EGFL8* expression correlates with increased patient survival, which could be due to its neuritogenic effect on peripheral neuroblastic tumor cells.

**The EGFL8 protein is significantly elevated in gang-lioneuromas compared to neuroblastomas and expressed by repair Schwann cells and stromal Schwann cells.** To verify whether the high *EGFL8* gene expression detected in GNs is reflected by the EGFL8 protein level, we performed high-resolution mass spectrometry analysis of SC stroma-rich GNs, SC stroma-poor NBs, as well as primary NB cultures and evaluated this data set together with our existing proteomic data set comprising repair SC-containing injured nerve tissue as well as primary repair-related SC cultures[17]. The results demonstrated a significantly higher abundance of the EGFL8 protein in injured nerves and GNs when compared to NBs (Fig. 7a). In addition, the protein levels of EGFL8 in primary cells matched their respective tissue of origin (Fig. 7a). This was confirmed by immunostaining illustrating SOX10 and EGFL8 co-expression by repair SCs in injured nerve tissue (Fig. 7b) and stromal SCs in GNs (Fig. 7c), while EGFL8 was absent on tumor cells in GN and NB primary

tumors (Fig. 7d,e). The images further showed that EGFL8 was also expressed in S100⁻ cells, e.g. in the perineurium and blood vessel-like structures (Fig. 7a,b).

Hence, mass spectrometric analyses and immunostaining confirmed that the EGFL8 protein is highly abundant in stromal SCs in GNs and repair SC in injured nerve tissues as well as primary repair-related SC cultures.

**EGFL8 protein is secreted by repair-related Schwann cells and rewires kinase-mediated signaling in neuroblastoma cells in vitro.** As EGFL8 is a predicted secreted factor and recombinant EGFL8 was able to induce neuronal differentiation, we next determined whether SC-produced EGFL8 is indeed secreted and investigated its mode-of-action in NB cells. First, we co-stained primary repair-related SC cultures for EGFL8 and membranous nerve growth factor receptor (NGFR), a marker associated with immature/repair SCs. EGFL8 showed an intracellular staining pattern with accumulation of positive signals in clusters of different sizes (Fig. 7f). 3D analysis illustrated EGFL8 positive vesicular structures embedded within the cytoplasm beneath the NGFR⁺ SC membrane (Supplementary Movie 1). In addition, we performed WB analysis for EGFL8 on cell lysates of human primary SCs, STA-NB-6 and SH-SY5Y cells, and conditioned culture medium (supernatants) of respective cultures (Fig. 7g, Supplementary Fig. 12). A GST-tagged recombinant EGFL8 protein was used as positive control. EGFL8 has an expected mass of 32 kDa, accordingly, the antibody detected the GST-tagged (GST corresponding to 26 kDa) recombinant EGFL8 protein at around 58 kDa in positive controls. In all four SC whole cell lysates, two bands were visible at around 32 and 37 kDa and three SC samples showed an additional band at around 55 kDa. In three out of four analyzed SC supernatants, prominent bands were detected at 37 kDa, which could indicate that the secreted EGFL8 protein underwent posttranslational modifications.

Second, we addressed the down-stream signaling of EGFL8 in NB primary cultures. As no data currently exist on EGFL8 receptor or signaling in human or any other mammalian cells, we employed an unbiased global- and phospho-proteomics approach in EGFL8-responsive STA-NB-6 versus non-responsive STA-NB-10 NB cells in a time-resolved manner. In total we identified 6385 and 6122 proteins expressed by STA-NB-6 and STA-NB-10 cells, respectively (Supplementary data 1). 6.408 and 6.133 sites were found phosphorylated corresponding to 1.851 and 1.820 proteins in STA-NB-6 and STA-NB-10, respectively (Supplementary Data 2). While STA-NB-6 showed a clear trajectory in the phospho-proteome upon 15 towards 60 min EGFL8 exposure (Supplementary Fig. 13a), a more diffuse dynamics was observed in STA-NB-10 (Supplementary Fig. 14a). As the most pronounced change was evident after 15 min, we focused on this time point and performed kinase enrichment analysis (KSEA), revealing a significant activation (enrichment z-score $\geq 1$, $p \leq 0.05$, substrate cutoff $\geq 3$) of 11 kinases in STA-NB-6 and 18 kinases in STA-NB-10 (Fig. 7h,

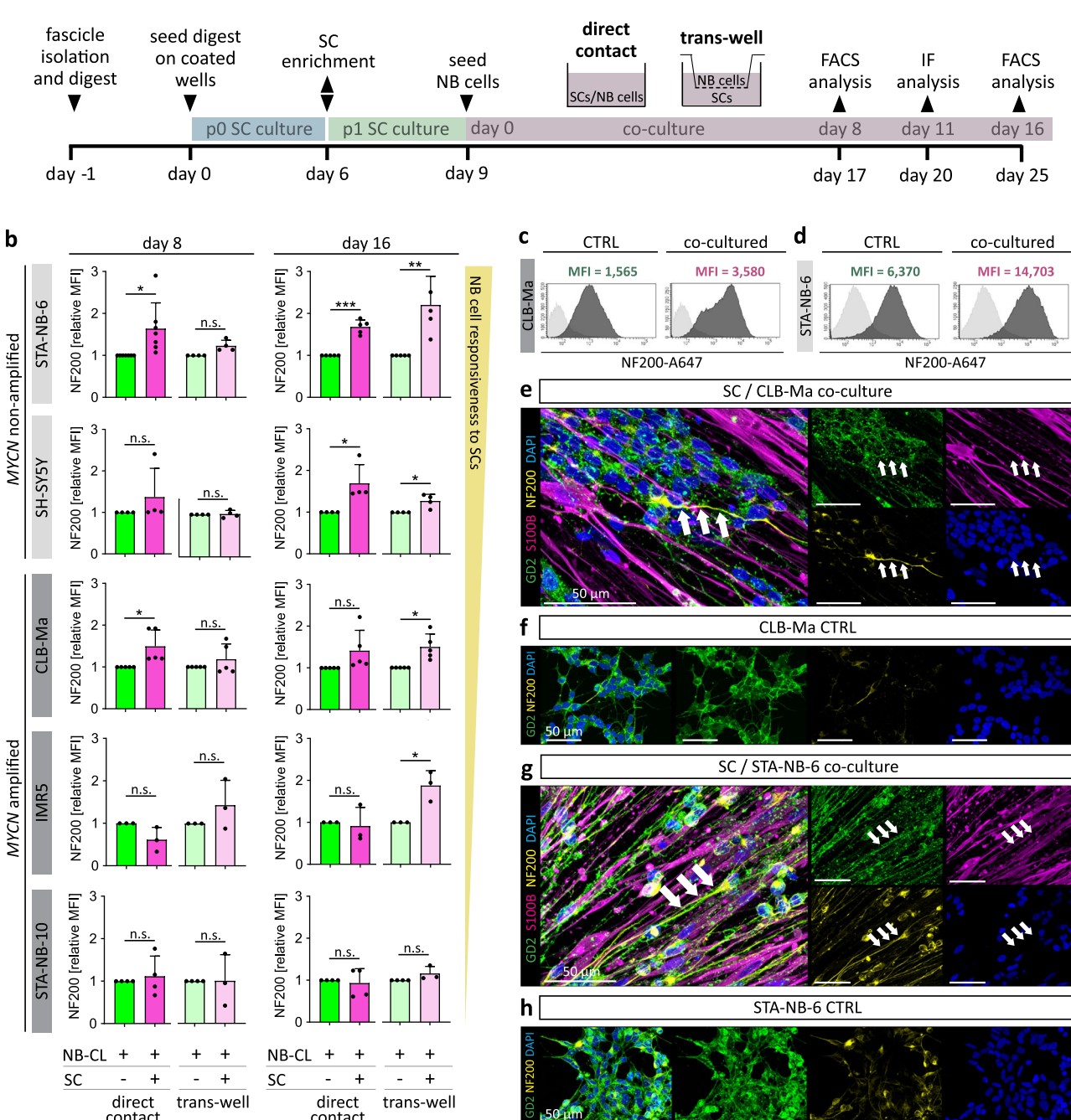

**Fig. 4 Neuronal differentiation analysis of neuroblastoma cell lines in response to repair-related SCs in vitro. a** Refined SC/NB cell co-culture set up including direct and trans-well co-cultures. Three NB cell lines and two NB cell short-term cultures were co-cultured with primary repair-related SCs and NF200 expression levels were analyzed by flow cytometry (FACS) and immunofluorescence (IF). **b** Bar diagrams show the normalized mean fluorescence intensity (MFI) of NF200 ± SD in GD2$^+$/S100B$^-$ NB cells upon direct co-cultures (STA-NB-6, $p = 0.033$; SH-SY5Y, $p = 0.363$; CLB-Ma, $p = 0.048$; IMR5, $p = 0.146$; STA-NB-10, $p = 0.666$) and trans-well co-cultures (STA-NB-6, $p = 0.042$; SH-SY5Y, $p = 0.816$; CLB-Ma, $p = 0.331$; IMR5, $p = 0.331$; STA-NB-10, $p = 0.988$) with SCs at day 8 as well as direct co-cultures (STA-NB-6, $p = 0.001$; SH-SY5Y, $p = 0.049$; CLB-Ma, $p = 0.132$; IMR5, $p = 0.762$; STA-NB-10, $p = 0.713$) and trans-well co-cultures (STA-NB-6, $p = 0.017$; SH-SY5Y, $p = 0.041$; CLB-Ma, $p = 0.023$; IMR5, $p = 0.050$; STA-NB-10, $p = 0.242$) with SCs at day 16. STA-NB-6: day 8 direct co-culture $n = 7$, trans-well $n = 4$, day 16 direct co-culture $n = 5$, trans-well $n = 4$; SH-SY5Y: $n = 4$; CLB-Ma: $n = 5$; IMR5: $n = 3$; STA-NB-10: direct co-culture $n = 4$; trans-well $n = 3$. A paired two-tailed Student's t-test comparing against the control was performed. $n$ refers to the number of independent experiments; * $p ≤ 0.05$; ** $p ≤ 0.01$; *** $p ≤ 0.001$; n.s. not significant. Representative FACS histograms show the unstained controls (light grey) and the MFI of NF200 in control and co-cultured (**c**) CLB-Ma and (**d**) STA-NB-6 cells (dark grey) at day 16. FACS gating strategy is detailed in Supplementary Fig. 7a. Representative IF images of co-cultured CLB-Ma cells (**e**) and STA-NB-6 (**g**) cells stained for NF200, S100B, GD2 and DAPI at day 11 of direct co-culture and respective NB cell controls stained for GD2, NF200, and DAPI (**f**, **h**); arrows indicate long neuritic processes of NB cells strongly positive for NF200 in co-cultures. Stainings were performed on NB cell controls and corresponding co-cultures with SCs derived from three independent donors.

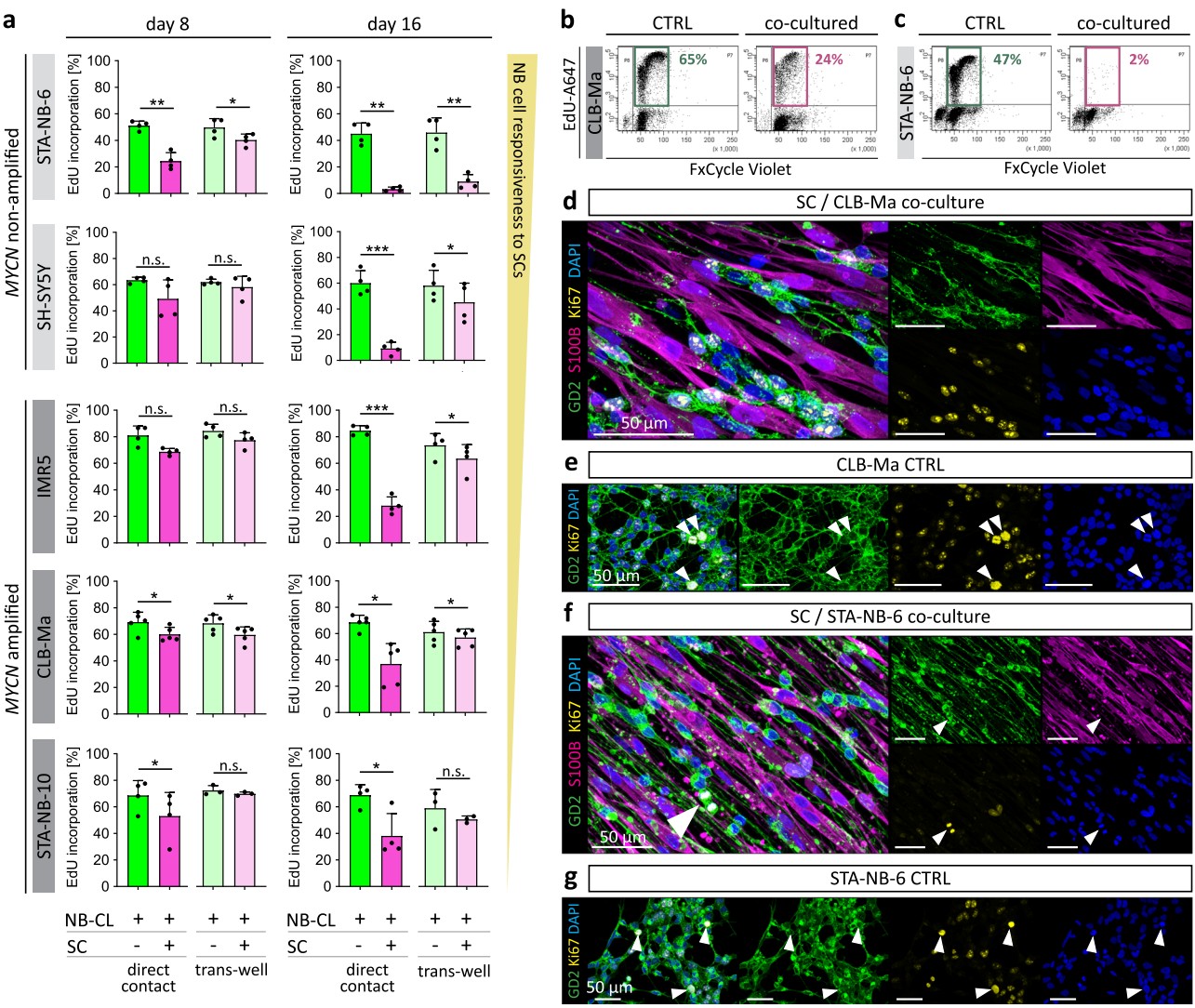

**Fig. 5 Proliferation analysis of neuroblastoma cell lines after direct and indirect contact to repair-related SCs in vitro.** Three NB cell lines and two NB cell short-term cultures were co-cultured with primary repair-related SCs and their proliferation rates were analyzed by FACS and IF. **a** Bar diagrams show the mean percentage of EdU-incorporation ± SD in GD2+/S100B- NB cells upon direct co-cultures (STA-NB-6, $p = 0.008$; SH-SY5Y, $p = 0.139$; CLB-Ma, $p = 0.025$; IMR5, $p = 0.073$; STA-NB-10, $p = 0.023$) and trans-well co-cultures (STA-NB-6, $p = 0.006$; SH-SY5Y, $p = 0.401$; CLB-Ma, $p = 0.032$; IMR5, $p = 0.051$; STA-NB-10, $p = 0.209$) with SCs at day 8 as well as direct co-cultures (STA-NB-6, $p = 0.003$; SH-SY5Y, $p = 0.001$; CLB-Ma, $p = 0.019$; IMR5, $p = 0.001$; STA-NB-10, $p = 0.037$) and trans-well co-cultures (STA-NB-6, $p = 0.015$; SH-SY5Y, $p = 0.042$; CLB-Ma, $p = 0.039$; IMR5, $p = 0.028$; STA-NB-10, $p = 0.472$) with SCs at day 16; STA-NB-6: $n = 4$; SH-SY5Y: $n = 4$; CLB-Ma: $n = 5$; IMR5: $n = 4$; STA-NB-10: direct co-cultures $n = 4$, trans-well cultures $n = 3$; A paired two-tailed Student's t-test comparing against the control was performed. * $p \leq 0.05$; ** $p \leq 0.01$; *** $p \leq 0.001$. Representative FACS plots illustrate EdU incorporation and the DNA content of control and co-cultured NB cells (**b**) CLB-Ma and (**c**) STA-NB-6 at day 16; the marked EdU+/FxCycleViolet+ cells are in the S-Phase of cell cycle. FACS gating strategy is detailed in Supplementary Fig. 7b. n refers to the number of independent experiments. Representative IF images of co-cultured CLB-Ma (**d**) and STA-NB-6 (**f**) cells stained for Ki67, S100B, GD2 and DAPI at day 11 of direct co-culture and respective NB cell controls stained for GD2, Ki67, and DAPI (**e**, **g**); arrows indicate NB cells undergoing mitosis. Stainings were performed on NB cell controls and corresponding co-cultures with SCs derived from three independent donors.

Supplementary Data 3, Supplementary Fig. 15). Kinases activated in STA-NB-6 and counter- or not regulated in STA-NB-10, such as HIPK1, p38β/MAPK11, ERK5/MAPK7, SGK1 and TLK2, and their substrates, e.g. PML, PAK2 or NDRG2, present key components of the EGFL8-induced signaling network (Fig. 7h, Supplementary Data 3 and 4, Supplementary Fig. 13-15).

In line with the predicted secretion of EGFL8, we here show that EGFL8 is present in vesicular structures within the cytoplasm and released in the medium of cultured human repair-related SCs. Further, we demonstrate that EGFL8 addition leads to a rapid (within 15 min) and specific phosphorylation of substrates of e.g. HIPK1, p38b/MAPK11, ERK5/MAPK7 only in EGFL8

responsive STA-NB-6, but not in the non-responsive STA-NB-10 short-term NB cell cultures, providing evidence for dynamic changes in the kinome associated with neuronal differentiation triggered by EGFL8.

## Discussion

This study presents a comparative analysis of human repair SCs in injured nerves and stromal SCs in GNs that builds upon previous efforts to delineate the role of SCs in nerve regeneration and the tumor microenvironment[1,17,31]. By investigating human tissues and primary cultures with deep RNA-sequencing, high-

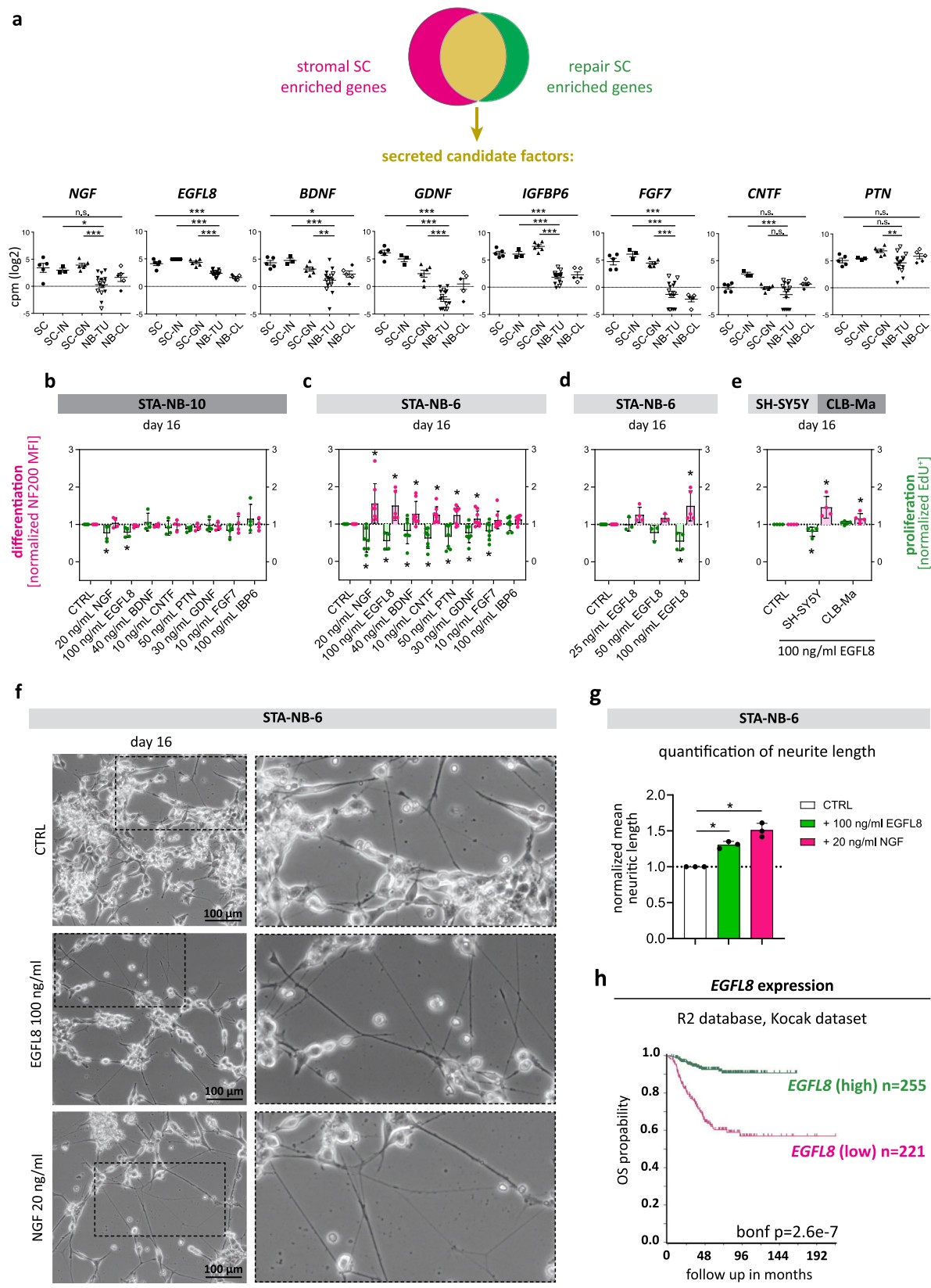

resolution mass spectrometry, and confocal imaging, we reveal a similar cellular state and functional competences of repair SCs and stromal SCs. Our comprehensive approach identified EGFL8 as a neuritogenic factor expressed by repair SCs and stromal SCs, which highlights matricellular proteins as tissue active components involved in regenerative and pathological responses of SCs

in the peripheral nervous system. Focusing on the interaction of tumor cells and SCs, we developed a co-culture model combined with a flow cytometry-based read-out demonstrating that NB cells react to repair-related SCs in a similar fashion as peripheral neurons upon injury. Moreover, the established co-culture model is broadly applicable and contributes to the ongoing research in

**Fig. 6 Neuronal differentiation and anti-proliferative effects of secreted factors shared by stromal and repair SCs. a** Expression levels of chosen candidate factors *NGF, EGFL8, BDNF, GDNF, IGFBP6, FGF7, CNTF* and *PTN* shown for primary repair-related SCs (SC, $n = 5$ biological replicates from 4 donors), repair SC rich injured nerve fascicle tissue (SC-IN, $n = 3$ biological replicates), SC stroma rich GN tissue (SC-GN, $n = 6$ biological replicates), NB tissue (NB-TU, $n = 15$ biological replicates) and NB short-term cell cultures (NB-CL, $n = 5$ biological replicates from 3 donors). Empty symbols indicate *MYCN* non-amplified NB-TUs and NB-CLs. Data are depicted as mean ± SD; *** $q \leq 0.001$, ** $q \leq 0.01$, * $q \leq 0.05$. **b–e** FACS analyses of neuronal differentiation (NF200 MFI) and proliferation (EdU incorporation) of NB cells treated with recombinant candidate proteins compared to untreated NB cell controls (CTRL) after 16 days of culture; data are shown as normalized mean values ± SD; n refers to the number of independent experiments; * $p \leq 0.05$. Proliferation levels (green) of (**b**) SC-weak responsive STA-NB-10 exposed to recombinant candidate proteins NGF ($p = 0.107$), EGFL8 ($p = 0.042$), BDNF ($p = 0.636$), CNTF ($p = 0.390$), PTN ($p = 0.026$), GDNF ($p = 0.671$), FGF7 ($p = 0.128$), and IBP6 ($p = 0.486$) compared to CTRLs (all $n = 4$) and (**c**) SC-strong responsive STA-NB-6 exposed to recombinant candidate proteins NGF ($p = 0.009$), EGFL8 ($p = 0.030$), BDNF ($p = 0.197$), CNTF ($p = 0.009$), PTN ($p = 0.033$), GDNF ($p = 0.042$), FGF7 ($p = 0.935$), and IBP6 ($p = 0.082$) compared to CTRLs (all $n = 7$, except EGFL8 $n = 4$) at concentrations as indicated. Differentiation levels (magenta) of (**b**) SC-weak responsive STA-NB-10 exposed to recombinant candidate proteins NGF ($p = 0.567$), EGFL8 ($p = 0.099$), BDNF ($p = 0.252$), CNTF ($p = 0.783$), PTN (0.153), GDNF ($p = 0.335$), FGF7 ($p = 0.934$), and IBP6 ($p = 0.926$) compared to CTRLs (all $n = 4$) and (**c**) SC-strong responsive STA-NB-6 exposed to recombinant candidate proteins NGF ($p = 0.022$), EGFL8 ($p = 0.049$), BDNF ($p = 0.050$), CNTF ($p = 0.014$), PTN ($p = 0.025$), GDNF ($p = 0.100$), FGF7 ($p = 0.015$), and IBP6 ($p = 0.232$) compared to CTRLs (all $n = 8$, except EGFL8 $n = 5$). **d** Proliferation levels (green) of STA-NB-6 cells exposed to 25ng/ml EGFL8 ($n = 3$, $p = 0.978$), 50 ng/ml EGFL8 ($n = 3$, $p = 0.196$), and 100 ng/ml EGFL8 ($n = 4$, $p = 0.030$) compared to CTRLs ($n = 4$). Differentiation levels (magenta) of STA-NB-6 cells exposed to 25ng/ml EGFL8 ($n = 4$, $p = 0.087$), 50 ng/ml EGFL8 ($n = 3$, $p = 0,106$), and 100 ng/ml EGFL8 ($n = 5$, $p = 0.049$) compared to CTRLs ($n = 5$). **b–e** Statistical test: One way ANOVA or mixed effects model and adjustments for multiple testing was performed. **e** Proliferation (green) and differentiation (magenta) levels of SH-SY5Y cells after treatment with 100 ng/ml EGFL8 (proliferation: $p = 0.050$, differentiation: $p = 0.050$) compared to CTRLs (both $n = 4$, paired, two-tailed Student's t-test) and CLB-Ma cells after treatment with 100 ng/ml EGFL8 (proliferation: $p = 0.116$, differentiation: $p = 0.046$) compared to CTRLs (both $n = 5$, paired, two-tailed Student's t-test); **f** Representative bright field images of STA-NB-6 cells at day 16 cultured in the absence (CTRL) or presence of 100 ng/ml EGFL8 or 20 ng/ml NGF ($n = 3$). Enlargements illustrate the neuritic processes of STA-NB-6 cells in CTRL as well as EGFL8- and NGF-treated cultures. **g** Quantification of neurite length of STA-NB-6 cells treated with 20 ng/ml NGF ($p = 0.004$) or 100 ng/ml EGFL8 (0.006) compared to untreated CTRLs. Data are depicted as normalized mean neurite length ± SD ($n = 6$ images per treatment over 3 independent biological replicates); Statistical test: repeated measures ANOVA and Dunnett's multiple comparison test; * $p$-value $\leq 0.05$. **h** Kaplan-Meier survival plot show the overall survival (OS) probability of patients grouped according to high and low *EGFL8* expression in primary tumors at diagnosis. Data were derived from the Kocak dataset (GSE45547) of the R2 Genomics Analysis and Visualization platform (https://r2.amc.nl) (see also Supplementary Fig. 11).

the field of regenerative medicine as well as cancer research aiming to elucidate the interplay of human SCs with different (tumor) cell populations.

The development of mostly benign-behaving peripheral neuroblastic tumors is hallmarked by an increasing stromal SC population and tumor cell differentiation along the sympathetic neuronal lineage. Since previous studies demonstrated that stromal SCs unlikely descend from tumor cells[1,28,30], we aimed to understand their origin and cellular state. The accumulation of publications supporting adult SCs as a highly plastic cell type urged us to investigate whether this reactive/adaptive potential plays a role in GNB/GN development.

The inherent SC plasticity is impressively demonstrated after peripheral nerve injury, where adult SCs undergo substantial expression and morphological changes to adapt their cellular functions to the needs of nerve repair[7,9,13,17]. In this study, transcriptome profiling elucidated a similar expression signature of stromal SCs in GNs and repair SCs in injured nerves as demonstrated by nerve repair-associated genes and functions. SC stroma development in peripheral neuroblastic tumors indeed exhibits parallels to the nerve injury-induced transformation of adult SCs into a repair cell identity, which is defined by two characteristics.

The first characteristic is the re-expression of genes associated with precursor/immature SCs that enables them to exit their differentiated cell state, re-enter the cell cycle, and gain an increased migratory capacity[10,50,51]. These features match the morphological observation of stromal SCs entering tumors through migration along blood vessels and connective tissue septa and the augmentation of SC stroma over time[28]. The expression of genes associated with a pre-myelin developmental stage of stromal SCs could also explain why the long axonal processes of ganglionic-like tumor cells in GNs are not myelinated[26].

The second characteristic of repair SCs is the acquisition of repair-specific functions including myelin clearance, macrophage recruitment, upregulation of MHC-II, formation of regeneration

tracks and expression of neurotrophic factors that support axon re-growth and guidance[7,9,14,16,17]. It is important to empathize that these competences are not shared by developing SCs or adult, i.e. differentiated, SCs. Previous studies identified the specific upregulation of ligands such as *GDNF* and *SHH* in repair SCs[12,13] and a recently published single-cell RNA-sequencing analysis comparing neonatal, uninjured (adult) and injured mouse peripheral nerves now provides further genes exclusively expressed by SCs upon injury. As the majority of them were present in the herein described enriched gene sets of human repair SCs and stromal SCs (see Fig. 2c), a contribution of mesenchymal stem cells or other precursors differentiating into Schwann-like stromal cells during GNB/GN development is unlikely.

Taken together, these findings assign the cellular state of stromal SCs to adult SCs that underwent a phenotypical switch as occurring after nerve damage and supports a repair-related cellular state of stromal SCs in GNs. Moreover, the progressing death of differentiating and differentiated neuroblastoma cells (ganglionic-like tumor cells) and resulting axon degeneration observed in GNBs/GNs[52] could supply stromal SCs with cues that trigger the repair-like state and explain why it does not diminish over time. As a consequence, stromal SCs could continuously exert nerve repair-associated functions in the microenvironment that are responsible for a benign tumor development.

Recognizing stromal SCs as possible facilitators of nerve repair-associated functions in the tumor microenvironment prompts the question how these functions could affect the behavior of tumor cells. We here show that stromal SCs share the expression of several neurotrophins and axon-guiding proteins with repair SCs in damaged nerves. Hence, neuronal differentiation-inducing cues derived from stromal SCs could be responsible for the differentiation of tumor cells into ganglionic-like cells during GNB/GN development. We modeled the interaction of SCs and NB cells in functional co-culture experiments, where we exposed genetically diverse NB cell lines and short-term primary NB cultures, derived from

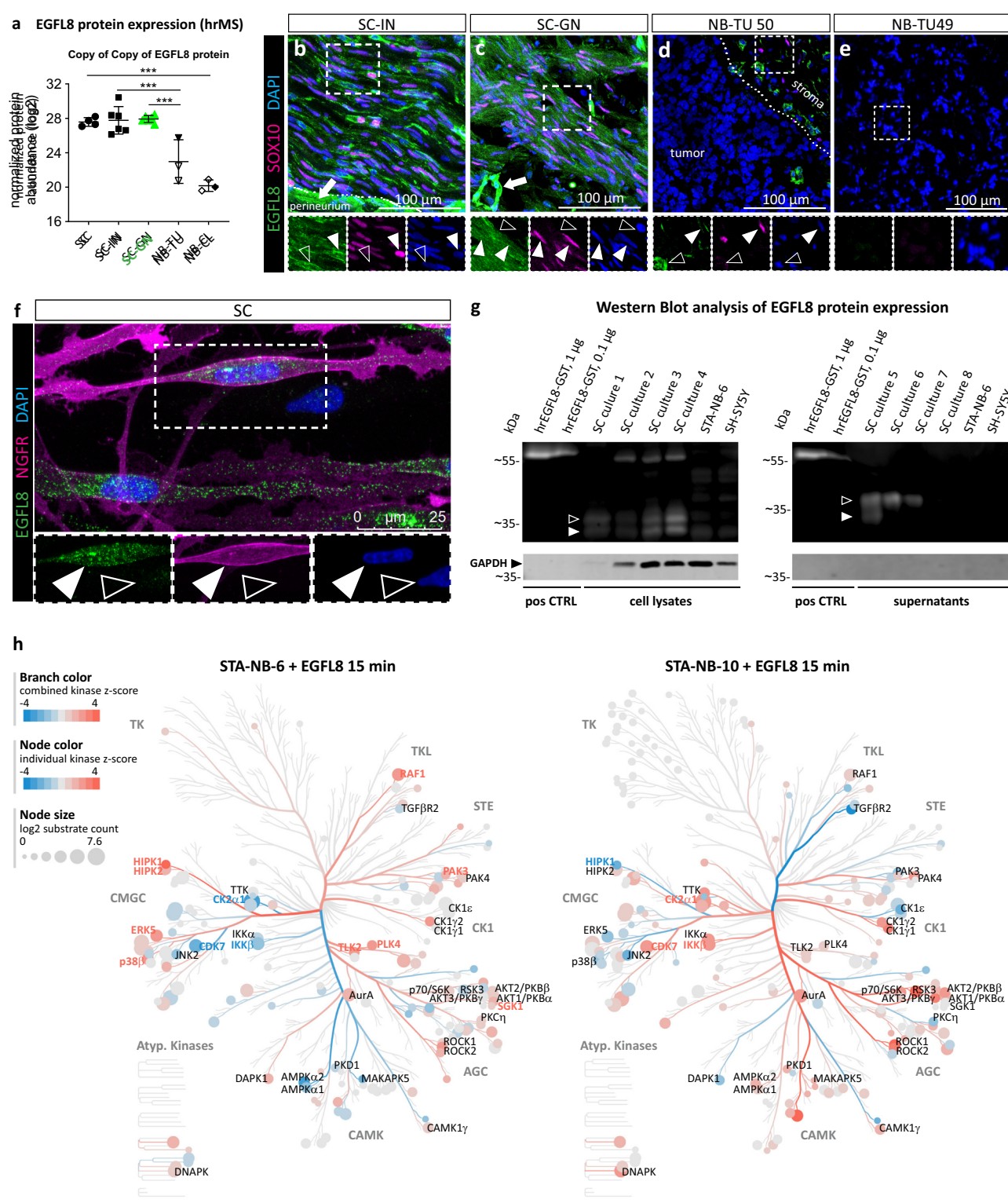

**a** EGFL8 protein expression (hrMS)

Copy of Copy of EGFL8 protein

(y-axis) normalized protein abundance (log2)

x-axis labels: SC, SC-IN, SC-GN, NB-TU, NB-EL

**b** SC-IN  **c** SC-GN  **d** NB-TU 50  **e** NB-TU49

EGFL8 SOX10 DAPI

perineurium   100 μm   100 μm   tumor / stroma  100 μm   100 μm

**f** SC

EGFL8 NGFR DAPI

0  μm  25

**g** Western Blot analysis of EGFL8 protein expression

pos CTRL | cell lysates    pos CTRL | supernatants

GAPDH

**h**

Branch color
combined kinase z-score
−4    4

Node color
individual kinase z-score
−4    4

Node size
log2 substrate count
0    7.6

STA-NB-6 + EGFL8 15 min

STA-NB-10 + EGFL8 15 min

aggressive high-risk NBs, to human primary repair-related SCs. Both, the direct contact to SCs and the in-direct contact to the SCs' secretome, were sufficient to induce neuronal differentiation and to impair proliferation of NB cells. Of note, this anti-tumor effect could be replicated by replacing SCs with recombinant neurotrophic factors discovered within the repair/stromal SC secretome.

In addition to their influence on neuroblastic tumor cells, stromal SCs also hold a considerable potential to modulate the tumor microenvironment. We found that stromal SCs express MHC-II, which is in line with other studies that reported the capacity of SCs to express MHC-II in (auto-) inflammatory or infectious neuropathies[53–56]. We also discovered that stromal SCs express potent chemokines and confirmed the presence of macrophages and T-cells in GNs, which is in accordance with the increasing reports about the immunomodulatory potential of SCs[17,56–61]. Furthermore, the shared expression signature of

**Fig. 7 EGFL8 protein expression analysis and kinome activation. a** High-resolution mass spectrometry (hrMS) data of EGFL8 protein expression levels in primary repair-related SCs (SC, $n = 4$ biological replicates), injured nerve tissue (SC-IN, $n = 6$ biological replicates from 3 donors), SC stroma-rich ganglioneuromas (SC-GN, $n = 6$ biological replicates), SC stroma-poor neuroblastomas (NB-TU, $n = 3$) and neuroblastoma short term cell cultures (NB-CL, $n = 3$); lined symbols indicate *MYCN* non-amplified NB-TUs and NB-CLs; *** $p \leq 0.001$; Data are depicted as mean ± SD. One-way ANOVA and Tukey's multi-comparison test was performed: SC vs. NB-TU adjusted $p = 0.0006$; SC vs. NB-CL $p < 0.0001$; SC-IN vs. NB-TU $p < 0.0001$; SC-IN vs. NB-CL $p < 0.0001$; SC-GN vs. NB-TU $p = 0.0004$; SC-GN vs. NB-CL $p < 0.0001$; Representative immunostaining of tissue cryosections of (**b**) SC-IN, (**c**) SC-GN, (**d**) NB-TU 50, and (**e**) NB-TU 49 stained for EGFL8, SOX10, and DAPI; filled arrowheads indicate EGFL8+/SOX10+ SCs, lined arrowheads indicate EGFL8−/SOX10− cells. Stainings were performed on three independent specimen per analyzed tissue. **f** Representative immunostaining of a repair-related SC culture stained for NGFR, EGFL8, and DAPI. The enlargement shows the SC body with membranous NGFR staining and intracellular EGFL8 signals within vesicle-like structures (filled arrowheads); the lined arrowheads indicate a NGFR−/EGFL8− fibroblast. A video of z-stacks visualizing the intracellular location of EGFL8 signals is available in Supplementary Movie 1. Stainings were performed on three independent SCs cultures. **g** Western blots show EGFL8 protein bands in primary repair-related SC lysates ($n = 4$) and supernatants ($n = 4$) but not in NB cell line STA-NB-6 and SH-SY5Y lysates and supernatants; filled white arrowheads indicate bands of about 32 kDa, the proposed molecular weight of EGFL8, lined white arrowheads indicate bands of ~37 kDa that could represent the EGFL8 protein with posttranscriptional modifications. Human recombinant (hr) EGFL8 (32 kDa) with a GST-tag (26 kDa) was used as positive control. Note that EFGL8 was detected via chemiluminescence and GAPDH via immunofluorescence on the same blot; full scans are available in Supplementary Fig. 12. **h** Phospho-proteomics upon 15 min EGFL8 treatment of EGFL8-responsive STA-NB-6 and non-responsive STA-NB-10 cells ($n = 3$ independent biological replicates). Kinase tree depicts kinases significantly enriched for substrate phosphorylation as compared to untreated control; for kinases with a cut-off ≥ 3 substrates; z-score ≥ 1; $p \leq 0.05$ kinase names are shown next to the node. For complete kinome trees see Supplementary Fig. 13b and 14b. Kinase families are labeled in grey; Kinases significantly enriched (red) or de-enriched (blue) in STA-NB-6 and not regulated or counter-regulated in STA-NB-10 are labeled in bold. Statistical test: kinase-substrate enrichment analysis of class 1 phosphosites ($p > 0.75$) utilizing PhosphoSitePlus and NetworKIN was performed, applying a NetworKIN score cutoff of 2, $p$-value cutoff of 0.05 and substrate count cutoff 3.

---

stromal/repair SCs contained basement membrane components and ECM remodelers such as metalloproteinases and matricellular proteins. Stromal SCs could therefore recruit and interact with immune cells, as well as execute tissue remodeling functions in the tumor environment with the original goal to rebuild an organized nerve structure similar to repair SCs upon nerve injury.

Taken together, the nerve repair-like phenotype equips stromal SCs with different strategies to influence their environment. Stromal SCs could either directly induce neuronal differentiation of peripheral neuroblastic tumor cells or indirectly manipulate the tumor microenvironment via immunomodulation and ECM remodeling responsible for a favorable tumor development.

Here, we introduce the matricellular protein EGFL8 as neuritogen. EGFL8 shares similar domains and molecular weight with EGFL7, which was described to induce neural stem cell differentiation[62,63]. We demonstrated high expression of the EGFL8 protein by repair SCs and stromal SCs and its secretion by repair-related SCs in vitro. Moreover, *EGFL8* expression in peripheral neuroblastic tumors correlated with an increased patient survival. We provide evidence for a neuritogenic function of human EGFL8, a protein of so far unknown function, as its recombinant form was sufficient to induce neuronal differentiation of NB cells at similar efficacy as NGF. Further, our comprehensive map of the activated kinome at baseline and upon EGFL8 stimulation delineates the down-stream signaling dynamics in NB cells. EGFL8 addition leads to specific phosphorylation of HIPK1, p38β/MAPK- and ERK5/MAPK7-substrates only in the sensitive cell line STA-NB-6, but not in the insensitive STA-NB-10. While ERK and MAPK are well established key nodes transmitting neurotrophic/neuritogenic signals[64], HIPK1, SGK1 and TLK2 have not been implicated in peripheral neuronal differentiation yet. Interestingly, the re-wiring of cellular signaling by EGFL8 converged at known regulators of neurogenesis, such as PML, NDRG2 and PAK2[65–67], corroborating the role of EGFL8 as neuritogenic factor. It will be interesting to elaborate the common and unique roles of EGFL8 in the concert of neurotrophic factors.

The discovery of EGFL8 as neuritogen underlines the increasingly recognized impact of matricellular proteins in injury response and pathological conditions[68–71]. Stromal SCs and repair SCs also shared the expression of other matricellular proteins such as *SPARC*, *SPP1* (osteopontin) and *CCN3* (*NOV*).

Notably, SC stroma-derived SPARC was previously reported to suppress NB progression by inhibiting angiogenesis and introducing changes in the ECM composition[72], suggesting that stromal/repair SCs are a source of various matricellular proteins that foster neuronal differentiation.

The plastic potential of adult SCs is a double-edged sword. While essential for nerve repair, recent studies point out its adverse effect in neuropathies and epithelial cancer progression[56,73]. Here, we demonstrate a favorable impact of SC plasticity on peripheral neuroblastic tumor cells as it manifests in SC stroma during the development of benignly behaving GNB/GN. The cellular similarities between stromal SCs and repair SCs suggest that stromal SCs are able to exert nerve repair-associated functions in the tumor microenvironment. Exploiting the strategies repair/stromal SCs use to generate a neuronal (re-)differentiation supporting environment could therefore hold a valuable therapeutic potential.

The prerequisite for a possible treatment approach is the susceptibility of aggressive NBs to SCs. We and others have previously investigated the effect of SCs and their secreted factors on aggressive NB cell lines. These studies confirmed that SCs are able to induce neuronal differentiation and impair the growth of NB cells, which were derived from SC-stroma poor high-risk NBs[31,32,34–36,74,75]. The confirmation that aggressive NB cells, although lacking the ability to attract SCs, are still responsive to SCs, offers essentially two therapeutic options[23], 1) including SC-derived factors as anti-tumor agents[1,76] and 2) the induction of SC stroma in aggressive NBs. Furthermore, identifying how the repair SC state can be sustained is also of high value for the field of regenerative medicine, since one of the main reasons for axonal regeneration failure after injury is the deterioration of repair SCs over time[46,77]. Thus, the more detailed knowledge about the molecular processes involved in GNB/GN development and nerve regeneration is promising to enrich treatment approaches for both nerve repair and aggressive NBs.

In conclusion, our study demonstrates that the cellular state of stromal SCs in GNs shares key features with repair SCs in injured nerves. This finding provides essential insight into GNB/GN development as it suggests that the inherent plasticity allows adult SCs to react to peripheral neuroblastic tumor cells in a similar way as to injured neurons. As a consequence, stromal SCs could exert repair-associated functions that shape an anti-tumor

microenvironment and induce neuronal differentiation of tumor cells responsible for a benign tumor behavior. Among the factors released by SCs, we identified the matricellular protein EGFL8 and report its neuritogenic effect on neuroblastic tumor cells. EGFL8 mediated neuronal differentiation through broad kinase activation including and beyond p38β/MAPK and ERK signaling, might hold considerable treatment possibilities for the therapy of aggressive NBs and patho-physiological conditions compromising peripheral nerve integrity.

## Methods

**Human material.** The collection and research use of human peripheral nerve tissues and human tumor specimen was conducted according to the guidelines of the Council for International Organizations of Medical Sciences (CIOMS) and World Health Organisation (WHO) and has been approved by the Ethikkommission Medizinische Universität Wien (EK2281/2016 and 1216/2018). Informed consent has been obtained from all patients or parents/guardians/legally authorized representatives participating in this study. The informed consent for obtaining peripheral nerve tissue covers the use of left over materials from medically indicated surgery for research purposes directed towards studying growth inhibition of aggressive neuroblastoma cells by human SC signals. The age-adapted informed consent for the CCRI Biobank covers the use of left over materials from medically necessary surgery or biopsy, which, after completion of routine diagnostic procedures, is biobanked (EK1853/2016) and available for research purposes, including genetic analysis, that are further specified in EK1216/2018: to conduct genetic, proteomic, imaging analysis and cell cultivation.

Neuroblastoma cell lines and primary cultures are available upon request. Primary Schwann cell cultures and tumor tissues are limited materials and therefore cannot be provided.

**Human peripheral nerve explants and primary Schwann cell cultures.** Human peripheral nerves were collected during reconstructive surgery, amputations or organ donations of male and female patients between 16 and 70 years of age. The ex vivo nerve injury model as well as the isolation procedure and culture conditions of primary human SCs have been performed as previously described[17,38]. Briefly, fascicles were pulled out of nerve explants and digested overnight using 1.25 U/ml Dispase II, 0.125% Collagenase type IV and 3mM calcium chloride. The fascicle-derived cell suspension was seeded on PLL/laminin coated dishes and cultured in SC expansion medium (SCEM: MEMα GlutaMAX™, 1% Pen/Strep, 1 mM sodium pyruvate, 25 mM HEPES, 10 ng/mL hu FGF basic, 10 ng/mL hu Heregulin-β1, 5 ng/mL hu PDGF-AA, 0.5% N2 supplement, 2 μM forskolin and 2% FCS. Cells of the initial seeding represent passage 0 (p0). Half of the medium was changed twice a week. When the cultures reached approx. 80% confluence, contaminating fibroblasts were depleted by exploiting their ability to adhere more rapidly to plastic. Enriched passage 1 (p1) SC cultures of about 96% purity, as determined via positivity for the SC marker S100B, were used for experimentation. For the ex vivo nerve injury model, about 1.5 cm long human nerve fascicles were subjected to an ex vivo degeneration period of 8 days in SCEM + 10% FCS at 37 °C (= injured nerve fascicle). During that time, axons degenerate and SCs adapt the repair phenotype within the explant[17]. Phase contrast microscopy images were generated using a Zeiss Axiovert 40C with the pixelink application version AL/A6XX.

**Neuroblastoma/ganglioneuroma tissue, neuroblastoma cell lines and patient-derived short-term cultures.** Tumor specimen from diagnostic NB tumors and GN tumors have been collected during surgery or biopsy for diagnostic purposes and left-overs were cryopreserved until analysis. Cryosections of GN tissue were analyzed for SC stroma rich areas identified by H+E-staining, immunofluorescence staining for SC marker S100B, and confirmed by a pathologist. The corresponding tumor region was excised using a scalpel blade and cryopreserved until RNA and protein extraction. The immunostaining analyses of cryosections were performed on three independent specimen per analyzed tissue, e.g. GN cryosections derived from three patients were stained for S100B, SOX10 and DAPI.

The used NB cell lines are derived from biopsies or surgical resection of aggressively behaving NB tumors of patients suffering from high-risk metastatic NBs. In-house established, short-term cultured primary NB cells STA-NB-6, -7 -10 and -15 as well as well-established NB cell lines SK-N-SH, SH-SY5Y, IMR5 and CLB-Ma were cultured in MEMα complete (MEMα GlutaMAX™, 1% Pen/Strep, 1 mM sodium pyruvate, 25 mM HEPES and 10% FCS). The NB cell lines, primary NB cultures and NB tumors differ in their genomic background including *MYCN*-amplification status. An overview of NB cell[78–84], primary NB cultures as well as NB and GN tumor characteristics is provided in Supplementary Tables 1 and 2.

**The co-culture model of primary Schwann cells and neuroblastoma cell lines or short-term cultured neuroblastoma cells.** NB cell lines or short term NB cultures (STA-NB-6, STA-NB-10, IMR5, SH-SY5Y and CLB-Ma) were co-cultured with enriched human p1 SCs from at least 3 independent donors. First, SCs were seeded in PLL/laminin coated wells of a 6-well plate in SCEM. At day 1 and day 2,

half of the media was exchanged with MEMα complete. At day 3, total media was changed to MEMα complete and NB cells were seeded directly to the p1 SC cultures as well as in PLL/laminin coated trans-wells (24 mm Inserts, 0.4 μm polyester membrane, COSTAR) placed above SC cultures, alongside with respective controls. Two third of the media was changed twice a week and one day prior to FACS analyses on day 8 and day 16.

For IF analysis, SCs from 3 independent donors were co-cultured with STA-NB-6 or CLB-Ma NB cell lines in coated wells of an 8-well chamber slide (Ibidi), respectively, alongside with controls for 11 days.

**Proliferation and differentiation FACS panels.** All antibody details are listed in Supplementary Table 3. If not stated otherwise, all steps of the staining procedures were performed on ice. The following antibodies have been conjugated to fluorochromes using commercially available kits according to the manufacturer´s instructions: anti-S100B has been conjugated to FITC (FLUKA) using Illustra NAP-5 columns (GE Healthcare), anti-GD2 (ch14:18, kindly provided by Professor Rupert Handgretinger, Department of Hematology/Oncology, Children's University Hospital, Tübingen, Germany) has been conjugated to AF546 using the AlexaFluor® 546 protein labeling kit (Molecular probes) and anti-NF200 has been conjugated to AF647 using the AlexaFluor® 647 protein labeling kit (Molecular probes).

Untreated SC and NB cell cultures, co-cultures or NB cultures stimulated with recombinant neuritogenic factors (Supplementary Table 3) were detached using Accutase (LifeTechnologies) and washed with FACS-buffer (1x PBS containing 0.1% BSA and 0.05% NaAzide). For the differentiation FACS panel, cells were incubated with GD2-AF546 for 20 min, washed once with FACS-buffer and fixed using Cytofix/Cytoperm (BD Biosciences) in the dark for 20 min. After washing with 1x perm/wash (BD), cells were stained with anti-S100B-FITC and NF200-A647 for 20 min. Cells were washed in 1x perm/wash and analyzed immediately at the FACSFortessa flow cytometer equipped with flow cytometer equipped with 5 lasers (355, 405, 488, 561 and 640 nm) and the FACSDiva software version 8.0 (both BD). For the proliferation FACS panel, 1 μM EdU was added to cultures for about 15 h. Cells were detached, washed and fixed in Roti-Histofix 4% for 20 min at RT. Permeabilization and EdU detection was carried out using the Click-iT EdU Alexa Fluor 647 Flow Cytometry Assay Kit (Thermo Fisher Scientific, C10419) according to the manufacturer's manual. Additional extracellular/intracellular staining was performed with GD2-A546 and anti-S100B-FITC antibodies in 1x saponin-based perm/wash for 30 min. After washing, cells were resuspended in 1x saponin-based perm/wash, 1 μl of FxCycle Violet (LifeTechnologies) DNA dye was added and samples were analyzed immediately at the FACSFortessa.

**Immunofluorescence staining and confocal image analysis.** All antibody details are listed in Supplementary Table 3. If not stated otherwise, the staining procedure was performed on RT and each washing step involved three washes with 1x PBS for 5 min. Primary antibodies against extracellular targets were diluted in 1x PBS containing 1% BSA and 1% serum; primary antibodies against intracellular targets were diluted in 1x PBS containing 1% BSA, 0.1% TritonX-100 and 1% serum. Briefly, thawed tissue cryosections or grown SC/NB cell co-cultures were fixed with Roti-Histofix 4% (ROTH) for 20 min at 4 °C, washed, and blocked with 1x PBS containing 1% BSA and 3% serum for 30 min. Cells and tissue sections were incubated with primary antibodies against extracellular targets, washed and incubated with appropriate secondary antibodies for 1h. Samples were then again fixed with Roti-Histofix 4% for 10 min. After washing, cells were permeabilized and blocked with 1x PBS containing 0.3% TritonX-100 and 3% serum for 10 min. When required, TUNEL staining was performed after permeabilization according to the manufacturer's protocol (PROMEGA). Samples were then incubated with primary antibodies against intracellular targets, washed and incubated with the appropriate secondary antibodies for 1h. Finally, samples were incubated with 2 μg/mL DAPI in 1x PBS for 2 min, washed and embedded in Fluoromount-G mounting medium (SouthernBiotech). Images were acquired with a confocal laser scanning microscope (Leica Microsystems, TCS SP8X) using Leica application suite X version 1.8.1.13759 or or LAS AF Lite version 4.0 software (Leica). Confocal images are shown as maximum projection of total z-stacks and brightness and contrast were adjusted in a homogenous manner using the Leica LAS AF software (Leica Microsystems).

**Quantification of neurite length and alignment.** The ImageJ plugin NeuronJ[85] was used to quantify the mean length of extended neurites by NB cells either on phase contrast images or immunofluorescence images between treated NB cells (co-culture with SCs, or exposure to EGFL8) and untreated NB cell controls (n = 3); at least two images were analyzed per condition. To evaluate the orientation (alignment) of NB cells after co-culture with SCs compared to NB cell cultured alone (n = 3), three GD2 stained immunofluorescence images per condition were analyzed with the ImageJ plugin OrientationJ *Measure* function [http://bigwww. epfl.ch/demo/orientation/], that calculates a distribution of pixels' orientations (varying from -90 to 90 degrees) per image. In order to merge information of all images per condition, the calculated distributions of orientations were mean-normalized resulting in a mean NB cell orientation of 0 degrees. To obtain a measure distinguishing the NB cell alignment between control and co-cultures, the

variance of the merged distributions was calculated (zero variance would reflect a perfect alignment). For each pair of measurements (control and co-culture), a Levene test[86] was applied to test for equal variances.

**RNA isolation, RNA sequencing and gene expression analysis.** Fresh frozen SC stroma-rich areas derived from diagnostic GNs (SC-GN, $n = 6$) were homogenized with the gentleMACS Dissociator (Miltenyi) using 1 mL of TRIzol per sample and the predefined RNA-01 gentleMACS program. RNA isolation was performed with the miRNeasy micro kit (Qiagen) following the manufacturer's protocol. Quantity and integrity of extracted RNA were assessed by the Qubit RNA HS Assay Kit (Life Technologies) and the Experion RNA StdSens Assay Kit (BioRad), respectively. 30 ng total RNA (RQI ≥ 8) was used for library preparation following the NEBNext Ultra RNA Library Prep Kit for Illumina protocol (New England BioLabs) with the Poly(A) mRNA Magnetic Isolation Module (New England BioLabs). After cDNA synthesis, the library was completed in an automated way at the EMBL Genomics Core Facility (Heidelberg, Germany). RNA-Seq was performed at the Illumina HiSeq 2000 platform and corresponding Illumna software (Illumina HiSeq Control software version 2.2.38, RTA version 1.18.61, HiSeq serial number HWI-ST999 and 50 bp-single-end reads were generated); basecalling was done with Illumina bcl2fastq-1.8.4.

The generated data were bioinformatically analyzed together with our previously published transcriptomic data sets of human primary SCs (SC, $n = 5$ samples from 4 donors), human injured fascicle explants (SC-IN, $n = 3$) and NB cells STA-NB-6 (analyzed in three biological replicates), STA-NB-7 and STA-NB-15 (NB-CL, $n = 6$ from 3 donors)[17], SC-rich areas of ganglioneuroma (SC-GN, $n = 6$) and diagnostic, untreated stage 4 NBs (NB-TU, $n = 15$)[87]. Respective GEO identifiers can be found in Supplementary Table 4.

Short read sequencing data was quality checked using FASTQC v0.11.5 (http://www.bioinformatics.babraham.ac.uk/projects/fastqc) and QoRTs v1.1.8[88] and then aligned to the human genome hs37d5 (ftp://ftp.1000genomes.ebi.ac.uk/) using the STAR aligner v2.5.3a[89] yielding a minimum of 11.6 million aligned reads in each sample. Further analysis was performed in R v3.4.1 statistical environment using Bioconductor v3.5 packages[90]. Count statistics for Ensembl (GRCh37.75) genes were obtained by the "featureCounts" function (package "Rsubread") and differential expression analysis was performed by edgeR and voom[91,92]. For differential gene expression analysis only genes passing a cpm (counts per gene per million reads in library) cut-off of 1 in more than two samples were included. All $p$-values were corrected for multiple testing by the Benjamini-Hochberg method. Genes with an adjusted $q$-value < 0.05 and a log2 fold change > 1 ($|\log2FC| > 1$) were referred to as 'significantly regulated' and used for functional annotation analysis via DAVID database[93].

**Western Blot analysis.** All antibody details are listed in Supplementary Table 3. Western blot analysis was performed as previously described[94,95]. 1x TBS-T was used for all washing steps that were performed three times for 5 min after each antibody incubation. Briefly, frozen cell aliquots were thawed, pelleted and lysed by addition of RIPA buffer. Culture media were centrifuged at 300 g for 10 min at 4 °C to remove cellular debris. The supernatants were mixed with −20 °C EtOH (1:5), precipitated at −20 °C for 20 h, centrifuged at 4000xg for 40 min at 4 °C and the dried pellet was lysed by addition of RIPA buffer. Protein extracts were stored in Protein LoBind tubes (Eppendorf) at −80 °C. Protein concentrations were determined via Bradford assay (BioRad). Protein extracts were mixed with SDS-loading buffer, denatured for 5 min at 95 °C, separated on a 10% SDS/PAA gel and blotted onto methanol-activated Amersham Hybond-P PVDF membranes. Membranes were blocked using 1x T-BST with 5% w/v nonfat dry milk for 30 min and incubated with anti-EGFL8 followed by HRP-conjugated secondary antibody. The blots were developed using the WesternBright Quantum detection kit (Advansta) and visualized with the FluorChemQ imaging system (Alpha Innotech, San Leandro, USA). Subsequently, membranes were incubated with anti-GAPDH followed by IRdye680T labeled secondary antibody. Blots were analyzed using the Odyssey imaging system (Licor) and the Odyssey software v3.0.

**Protein isolation, high-resolution mass spectrometry and expression analysis.** Fresh frozen diagnostic GN-derived SC stroma-rich areas (SC-GN, $n = 6$), diagnostic high-risk NB tumors (NB-TU, $n = 3$) as well as low-passage NB cells STA-NB-7, STA-NB-2 and STA-NB-10 (NB-CL, $n = 3$) were used for proteomic analysis (see Supplementary Table 1 & 2 for tumor and cell line characteristics). Protein isolation from cells and tissue, mass spectrometry sample preparation and liquid chromatography-mass spectrometry (LC-MS) has been carried out[17,96], all samples were measured in two technical replicates. LC-MS/MS analyses were performed using a Dionex Ultimate 3000 nano LC-system coupled to a QExactive orbitrap mass spectrometer with software TUNE version 2.5-204201/2.5.0.2042 and Chromeleon version 6.0 (all Thermo Fisher Scientific).

For the identification and label free quantification of proteins, the MaxQuant software package (version 1.6.1.0)[97] was used[98]. The human UniProt database (version 03/2018, restricted to reviewed entries only) with 20316 entries was used for the search, and the false discovery rate (FDR) was set to 0.01 on both peptide and protein level. The alignment time window was set to 1 min, with a match time window of 5 min. The four data matrices obtained as described above were loaded

into Perseus software (version 1.6.7.0), followed by filtering those analytes that were present in at least 70% of samples in at least one group[99]. Next, data were log 2 transformed, and missing values were replaced by normally distributed random numbers with a set width of 0.3 and a downshift of 1.8. A two sided t-test was applied for statistical significance testing with number of randomizations set to 250, the FDR threshold was set to 0.05 and the S0 value to 0.1. The data were analysed together with previously generated proteomic data set of human primary SCs and human injured fascicle explants[17].

**Phosphoproteomics.** EGFL8-treated (100 ng/mL) and untreated cell lines were lysed with 4% SDC buffer containing 100 mM Tris-HCL (pH 8.5) to the cells. The lysate was collected and heat-treated at 95 °C for 5 min. Three biological replicates were performed for each cell line.

For the phosphopeptide enrichment, a slightly modified protocol of the EasyPhos workflow was applied[100]. Briefly, a total of 200 μg protein was used for the enrichment procedure. Protein reduction using 100 mM TCEP and alkylation using 400 mM 2-CAM with subsequent enzymatic digestion with Trypsin/Lys-C mixture (1:100 Enzyme to Substrate ratio) at 37 °C for 18 h was performed. The solution containing the peptides was mixed with enrichment buffer containing 48% TFA (vol/vol) and 8mM Potassium dihydrogen phosphate. Samples were incubated with 3mg TiO2 Titansphere beads (GL Sciences) for 5 min at 40 °C with subsequent washing and elution from StageTips with 40% ACN and 5% Ammonium hydroxide solution. Samples were dried and reconstituted in 15 μL MS loading buffer containing 97.7% H2O, 2% ACN and 0.3% TFA.

For the global proteome, a digestion protocol S-trap technology was employed[101]. In short, proteins were solubilized in buffer containing 5% SDS with subsequent reduction and alkylation using 64 mM DTT and 48 mM IAA, respectively. After addition of trapping buffer (90% vol/vol methanol, 0.1 M triethylammonium bicarbonate) samples were loaded onto S-trap cartridges and digested with Trypsin/Lys-C Mix at 37 °C for 2 h. Supernatants containing the collected peptides were dried. Dried peptide samples were reconstituted in 5 μL 30% formic acid (FA) containing 10 fmol of 4 synthetic standard peptides each and diluted with 40 μL mobile phase A (99.9% H2O, 0.1% FA). LC-MS/MS analyses were performed using a Dionex Ultimate 3000 nano LC-system (Thermo Fisher Scientific) coupled to a timsTOF pro mass spectrometer (Bruker Daltonics). 10 and 5 μL of phosphopeptide enriched and global proteome samples, respectively, were loaded on a 2 cm x 100μm C18 Pepmap100 pre-column (Thermo Fisher Scientific) at a flow rate of 10 μL/min using mobile phase A. Afterwards, peptides were eluted from the pre-column to a 25 cm x 75 μm 25cm Aurora Series emitter column (Ionopticks) at a flow rate of 300 nL/min and separation was achieved using a gradient of 8% to 40% mobile phase B (79.9% acetonitrile, 20% H2O, 0.1% FA) over 90 min.

Data analysis was performed using MaxQuant 1.6.17.0[97] employing the Andromeda search engine was used for protein identification against the UniProt Database (version 12/2019 with 20 380 entries) allowing a mass tolerance of 20ppm for MS spectra and 40ppm for MS/MS spectra, a FDR < 0.01 and a maximum of 2 missed cleavages. Furthermore, search criteria included carbamidomethylation of cysteine as fixed modification and methionine oxidation, N-terminal protein acetylation as well as phosphorylation of serine, threonine and tyrosine as variable modifications. For the interpretation of phosphoproteomics data, a kinase-substrate enrichment analysis of class 1 phosphosites ($p > 0.75$) utilizing PhosphoSitePlus and NetworKIN was performed, applying a NetworKIN score cutoff of 2, $p$-value cutoff of 0.05 and substrate count cutoff 3[102–104]. For the visualization of enriched kinases in context of the global kinome, the application Coral was used[105].

**Statistical analyses.** If not mentioned otherwise, Excel 2016 and GraphPad Prism 8 was used for statistical analysis. Values were given as means ± SD of at least 3 independent biological samples or independent biological replicates. For paired analyses a Student's t-test, for parametric analysis of multiple conditions one-way ANOVA and Tukey's multiple comparisons post-hoc test was performed. $p$-values ≤ 0.05 were considered significant.

**Reporting Summary.** Further information on research design is available in the Nature Research Reporting Summary linked to this article.

## Data availability

All data sets produced and used in this study are available in public repositories as listed in Supplementary Table 4. RNA-sequencing datasets were uploaded to the gene expression omnibus (GEO) repository (https://www.ncbi.nlm.nih.gov/geo/) with the dataset identifiers GSE90711, GSE94035, GSE147635, the Kocak dataset GSE45547 and NRC dataset GSE85047 are publicly available. The mass spectrometry global and phosphospho-proteomics data have been deposited to the ProteomeXchange Consortium (http://proteomecentral.proteomexchange.org) via the PRIDE partner repository[106] with the dataset identifier PXD018267 and PXD022217 and are publicly available. Source data are provided with this paper.

## Code availability

No custom codes have been developed in this study.

# ARTICLE

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

## Acknowledgements

This study was supported by Österreichische Forschungsförderungsgesellschaft (FFG) grants (ID:844198, TisQuant, EraSME, by the Austrian Research Promotion Agency, to P.F. Ambros and 10959423, VISIOMICS, Coin Netzwerke, to S. Taschner-Mandl), the European Union's Seventh Framework Program (FP7/2007–2013) under the project ENCCA, grant agreement HEALTH-F2-2011-261474, the Herzfeldersche Familien-stiftung, Modicell (MC-IAPP Project 285875) and St. Anna Kinderkrebsforschung. We thank Helmut Dolznig (Institute of Medical Genetics, Medical University of Vienna), Rudolf Oehler and Stephan Zeindl (Department of Surgery and Comprehensive Cancer Center, Medical University of Vienna) for providing human immortalized fibroblasts and cancer associated fibroblasts. We are also grateful to Ulrike Pötschger (Children's Cancer Research Institute) for advice regarding statistical analysis.

## Author contributions

T.W. and S.T.-M. planned experiments, performed research, analyzed and interpreted data and wrote the manuscript; H.S., A.B., L.J. and F.R. performed research and analyzed data; F.K. analyzed data; C.F. and M.K. developed bioinformatics tools and analyzed data; C.G. analyzed and interpreted data; R.W. provided essential material; P.F.A., I.M.A., and S.T.-M. conceptualized the project; P.F.A. and I.M.A interpreted data; all authors reviewed the manuscript.

## Competing interests

The authors declare no conflict of interest.
