## [Peer Review File · Nature Communications]

Reviewer #1 (Remarks to the Author):

In the manuscript entitled “Schwann cell plasticity regulates neuroblastic tumor cell differentiation via epidermal growth factor- like protein 8” by Weiss, Taschner Mandl et al. present a transcriptomics-driven study of the beneficial role of Schwann cells in benign tumors of the peripheral nervous system, leading to insight into the potential of Schwann-cell derived soluble factors to prevent or treat malignancies.

Starting from the notion that Schwann cells of supposedly non-tumor origin found in benign ganglioneuromas and absent from usually malign neuroblastomas might be a causative factor to the fate of these tumors, the authors performed RNA sequencing of patient samples of the tumor types along with injured nerves, isolated primary Schwann cells and neuroblastoma cell lines. A comparison of the transcriptomes revealed a repair-associated signature in ganglioneuromas resembling that of injured nerves and primary Schwann cells. As expected, as they are devoid of Schwann cells, the neuroblastoma samples clustered farther away from the Schwann-cell containing samples. To then validate their concept of the role of Schwann cells in benign tumors and to evaluate a putative beneficial effect of Schwann cells on malignant tumors, the authors performed co-culture of primary Schwann cells with neuroblastoma cell lines. Indeed, the co-cultured neurons showed increased differentiation markers as well as a decrease in proliferation and an increase in apoptosis.

Importantly, the potentially beneficial effects on neuronal tumor cells were also present without direct contact of neurons and co-cultured Schwann cells. The authors therefore extracted potentially secreted factors from the RNAseq data and chose to test one of these factors, epidermal growth factor- like protein 8 (EGFL8), for effects on neuroblastoma cell lines. Treatment with recombinant EGFL8 resulted in concentration dependent pro-differentiating and anti-proliferating effects that were comparable with those of nerve growth factor. In line with a role of EGFL8 as a Schwann-cell secreted anti-tumor agent, EGFL8 protein was detected in supernatants of Schwann cell cultures, and high EGFL8 protein levels were measured in patient ganglioneuromas but not in neuroblastomas.

This is a compelling study that is well performed. Most of the conclusions are soundly supported by data. The RNAseq data was adequately analysed.

The discovery of the roles of (repair) Schwann cells and secreted EGFL8 for tumor development is of utmost importance for both the cancer field and the glial physiology field. I find the manuscript clearly and well written, and I recommend it for publication after minor changes.

Minor comments:

- The authors stress a novel, neuritogenic function of EGFL8 in several instances of the manuscript. However, I find the evidence presented not sufficient to support this claim. I see that neuroblastoma cells orient themselves and also project neurites along co-cultured Schwann cells (Fig. 3). Whether this effect is brought about in co-culture without direct contact or via EGFL8 treatment is not evident from the data. Fig. 6 does indicate an increase in amount and length of neurites after EGFL8 treatment, however these parameters would have to be quantified in order to provide evidence for EGFL8 as a neuritogen. The authors should either do this or refrain from this conclusion, which is not essential for the overall significance of the paper.

- There are a number of typos to correct, the authors should use a possible revision to thoroughly proof-read the text. Examples are found on p.6,l.18 (neuritogenic instead of neuritogenic) and l.25

(ganglioneuroblastomas instead of ganlgioneuroblastomas).

Reviewer #2 (Remarks to the Author):

In this paper Weiss et al. explore the role of stromal cells in ganglioneuroma and neuroblastoma. They propose and begin to validate the interesting hypothesis that stromal cells resemble repair Schwann cells, and they promote more benign evolution of neural tumors. By comparing transcriptomic profile analysis of neuroblastoma (which have few stromal cells), ganglioneuroma (rich in stromal cells) and repair Schwann cells in injured nerves, the authors identify a common signature of stromal cells in ganglioneuroma and repair Schwann cells in injured nerve. They next analyze the secretome of repair Schwann cells, and among various factors they show that EGFL8 promote differentiation and decrease proliferation of a neuroblastoma tumor cell line. Interestingly, EGFL8 may also expressed by stromal cells. Overall the data are interesting and novel, and mostly convincing. This study uses human tissue and human cell lines and has direct clinical relevance. Moreover, the authors show an impressive correlation of tumor EGFL8 expression with patient survival rate. However the paper also suffers from some experimental weaknesses, that include lack of crucial quantifications and of some controls. These should be addressed to make the results stronger:

- the transcriptomic analysis is interesting, but given the complexity of the cellular composition of the different starting tissues, the data may be more difficult to interpret than the authors describe. Isolation of stromal cells and analysis of their transcriptome, or validation with single cell transcriptomic would strengthen the results. At minimum, confirmation of expression of EGFL8 by stromal cells (isolated, or in immunohistochemistry) would be essential.
- Figure 1 A-C requires quantification.
- Figure 3. The alignment and increase neurite outgrowth should be quantified. Can stromal cells be isolated and cultured, to show that they have the same effect of human Schwann cells, for example on the neuroblastoma alignment shown in figure 3?
- absence of GD2 expression in Schwann cells in figure 3F should be confirmed and shown.
- Previous papers addressed the role of Schwann cells in neuroblastoma (e.g. Cross-talk between Schwann cells and neuroblasts influences the biology of neuroblastoma xenografts. Liu S, Tian Y, Chlenski A, Yang Q, Zage P, Salwen HR, Crawford SE, Cohn SL. Am J Pathol. 2005 Mar;166(3):891-900). These papers should be cited and detract a bit from the novelty.
- Figures 3, 4, 5, and throughout the manuscript: the authors frequently refer to the primary Schwann cells as repair Schwann cells. While primary Schwann cells share many features with repair Schwann cells, it is possible that these cells do not truly remain in the repair Schwann cell state once passaged in culture. For clarity, the authors could use the term primary Schwann cells to refer to these cells throughout the manuscript.
- Figure 4B, the number of neurofilament positive cells should be quantified

- Jun is one of the key markers of repair Schwann cells. Immunohistochemistry showing JUN staining in SC-GN tissue would further validate the authors' transcriptomic findings.
- Olig1 and Shh are also key markers used to distinguish repair Schwann cells from other Schwann cell subtypes (Jessen and Mirsky, 2019, author's reference number 63). Did the authors examine these markers in their transcriptomic profiling of repair Schwann cells and stromal Schwann cells?
- Fig. S1: the number of T cells present in neuroblastoma or human injured nerves should be checked.
- Is EGFL8 necessary and sufficient? would silencing of EGFL8 in stromal or Schwann cells diminish the effect of neuroblastoma cells?

Minor points:

Figure 1 – For completeness and to demonstrate similarity of neuroblastoma cell lines to patient neuroblastoma tissue, could the authors also show immunostains of GD2 in ganglioneuroma and neuroblastoma tissue?

The ability of repair SCs to take up GD2 (figure 3f) is an interesting finding. Could the authors comment further on this and whether (speculatively) stromal Schwann cells might act in a phagocytic manner in GN or NB tissue?

It is interesting that the transwell (i.e. indirect contact) seems to have a greater effect on NB differentiation in CLB-Ma and IMRS cell lines (figure 4b), yet direct contact has a greater effect on proliferation in these cell lines (figure 5b). Could the authors comment on whether they think this might be biologically relevant?

The role and expression of HLA in Schwann cells has been extensively documented by many papers, some of them just from a quick search are listed below. HLA expression by Schwann cells is not so novel and some of these papers should be referenced.

Page 3, third bullet point – “capable to induce neuronal differentiation” reads better as “capable of inducing neuronal differentiation”

Page 4, abbreviations section – typo “ganglioneuroblatoma” should be “ganglioneuroblastoma”

Page 6 – typo “ganlgioneuroblastomas” should be “ganglioneuroblastomas”

Page 9 – in the text, could the authors clarify that the active protein modification and transport machinery is in repair SCs, rather than stromal SCs?

Page 10 – “alongside (with) controls for 11 days”; could the authors state clearly in the text what the controls were (i.e. Schwann cells or NBT cells alone). Also this phrase might read more easily as “alongside controls for 11 days”.

Page 13/ figure 6g – could the authors be clear as to why the human data sets comprise 498 (presumably corresponding to supp fig 4?) and 649 tumor specimens, but only 471 patient survival outcomes are reported in 6g?

Page 14 – authors switch from using “vimentin” to “vime” but should be consistent throughout the manuscript

Figure 1 – typo in 1b bahavior should be behavior

Figure 1 legend – please could the authors define the acronym MNA

Figure 1 legend – the n numbers reported in the legend for RNA seq do not appear to match up with the data represented in the figure

Figure 3 – could the authors include a low magnification panel of SC/CLB-Ma co-cultures, similar to in 3d?

Figure 3 legend – typo (b) GD2+/vimentin~ should be (b) GD2+/vimentin+

Figure 7 – could the authors include a video of confocal Z stacks to further illustrate the subcellular localization of EGFL8 vesicular structures?

Figure 7D – could the authors add arrows to the blots to indicate the EGFL8 bands?

Figure 7D – typo third lane heading of first blot and all headings of second blot: “culure” should be “culture”

Figure 7D – while I appreciate that this may have been a difficult antibody to work with, the bands on the Western blots are not crisp; do the authors have better example blots they could use in this figure?

Materials and methods pg 20 – define FBs

Materials and methods pg 21 – supplementary tables 1 and 2 show NB and GN tumor characteristics, supplementary table 3 is antibodies

HLA in Schwann cell references:

Expression of antigen processing and presenting molecules by Schwann cells in inflammatory neuropathies. Meyer Zu Horste G, Heidenreich H, Lehmann HC, Ferrone S, Hartung HP, Wiendl H, Kieseier BC. *Glia*. 2010 Jan 1;58(1):80-92. doi: 10.1002/glia.20903.

Immunohistochemical localizations of class II antigens and nerve fibers in human carious teeth: HLA-DR immunoreactivity in Schwann cells. Yoshida N, Yoshida K, Iwaku M, Ozawa H. *Arch Histol Cytol*. 1998 Oct;61(4):343-52.

Axonal neuropathy associated with interferon-alpha treatment for hepatitis C: HLA-DR immunoreactivity in Schwann cells. Quattrini A, Comi G, Nemni R, Martinelli V, Villa A, Caimi M, Wrabetz L, Canal N. *Acta Neuropathol.* 1997 Nov;94(5):504-8.

LA-DR expression in peripheral neuropathies: the role of Schwann cells, resident and hematogenous macrophages, and endoneurial fibroblasts. Sommer C, Schröder JM. *Acta Neuropathol.* 1995;89(1):63-71

Class II antigen expression in peripheral neuropathies. Mitchell GW, Williams GS, Bosch EP, Hart MN. *J Neurol Sci.* 1991 Apr;102(2):170-6.

Class II antigen expression on human cultured Schwann cells from patients with Charcot-Marie-Tooth disease. De Martini I, Bianchini D, Schenone A, Cadoni A, Zicca A, Zaccheo D, Mancardi GL. *Neurosci Lett.* 1989 May 22;100(1-3):331-4.

HLA-DR Schwann cell reactivity in peripheral neuropathies of different origins. Mancardi GL, Cadoni A, Zicca A, Schenone A, Tabaton M, De Martini I, Zaccheo D. *Neurology.* 1988 Jun;38(6):848-51.

HLA-DR-expressing cells and T-lymphocytes in sural nerve biopsies. Schrøder HD, Olsson T, Solders G, Kristensson K, Link H. *Muscle Nerve.* 1988 Aug;11(8):864-70.

Reviewer #3 (Remarks to the Author):

The study of Weiss and colleagues starts from the observation that ganglioneuromas, which contain a Schwann cell-rich stroma, are more benign than related neuroblastomas, which have very low numbers of Schwann cells. To study Schwann cell properties and their role in these tumors, the authors generated expression profiles, compared them with expression profiles of injured nerves and concluded that stromal Schwann cells are very similar to repair Schwann cells in the nerve. They went on to show that primary Schwann cells stimulate differentiation, inhibit proliferation and induce cell death in several neuroblastoma cell lines, and that Schwann-cell produced and secreted EGFL8 is responsible for at least part of the pro-differentiating effect. This leads them to conclude that the repair phenotype allows Schwann cells to influence tumor cell behavior. The study is interesting in principle and yields insights into the modulatory role of Schwann cells in neural crest-derived tumors. The manuscript is professionally put together and well written. Data are clearly presented. Still I find several issues problematic.

In their RNA-Seq studies, the authors generate expression profiles of tumors and injured nerve, then subtract neuroblastoma-expressed genes from those expressed in ganglioneuromas or injured nerve, and compared the two gene sets. Considering that the biggest difference between ganglioneuromas and neuroblastomas are Schwann cells that also make up a large portion of the cells in the injured nerve, it is almost a self-fulfilling prophecy that the authors detect a common Schwann cell-specific signature in both analyzed groups. Whether this common signature is that of a repair Schwann cell is, however, not fully substantiated. From the absence of myelin in the tumors, it

is obvious that the Schwann cells within the tumor are not myelinating Schwann cells, but whether they are really repair Schwann cells or some other Schwann cell type (such as immature Schwann cells that are present during ontogenetic development) is difficult to judge. In support of their assumption, the authors point out that the signature of stromal Schwann cells resembles that of primary cultured Schwann cells, which they claim to closely resemble repair Schwann cells. They even go as far as referring to cultured primary Schwann cells as cultured repair Schwann cells. To my mind this is not justified. It is obvious that Schwann cells in cell culture have a number of properties that they share with repair Schwann cells. However, they share these same properties with all other types of proliferative Schwann cells. This may all seem like semantics. However, these semantics are important for this paper, because the fundamental claim of the paper builds on it.

For analyzing the effect of Schwann cells on tumor cells (Figs. 4,5; S.Fig. 2), the authors exclusively rely on neuroblastoma cell lines (whose response varies substantially). Long established neuroblastoma cell lines have adjusted to growth in culture and may substantially differ in their response from primary tumor cells. This has not been taken into consideration. As carried out there is furthermore no evidence that the observed pro-differentiating, anti-proliferating and pro-apoptotic effects are specific to Schwann cells. How can the authors exclude that fibroblasts (or any other cell type) do not have the same effects when co-cultured with neuroblastoma cells?

Among the Schwann cell effects on tumor cells, the pro-differentiating effect is probably the most specific (Fig. 4, see above). When looking at the data, it is also the least robust. Therefore, it would be reassuring to see the pro-differentiating effect confirmed in immunocytochemical stainings with several markers. In the current state, most immunocytochemical analyses throughout the paper are carried out with a very limited set of markers, mostly NF200 as neuronal marker and S100B as Schwann cell marker.

Mechanistic data are missing on the molecular mode of EGFL8 action.

Other issues:

Fig.1a and p.8: The authors state that degrading axons in injured nerves are identified by NF200 staining. NF200 labels nerves, not specifically degrading ones.

Fig.2: The authors argue that that the common signature obtained from comparing tumors and injured nerves is a Schwann cell signature. While this is true for the most part, some aspects of the signature are also contributed by immune cells that are present in injured nerves and differentially occur in ganglioneuromas vs. neuroblastomas. Therefore the authors cannot formally exclude that terms associated to immune regulation, phagocytosis etc. are enriched in the signature because of immune rather than Schwann cells. Further experiments are needed.

Fig. 3d,e: These experiments additionally require a quantitative read-out.

Fig. 7d: What is the 55 kDa band in some of the Schwann cell lysates. It is difficult to judge whether the approx. 35 kD band in lysates and supernatants is the same, because samples are on different gels with different running behavior.

There is no evidence for a role of EGFL8 on nerve regeneration. This would be an important piece of

data as the authors imply similar function of the same cell type in injury and tumors. However, I understand that this may be beyond the scope of the study.

Vienna, Oct 30th 2020

#dear Editor,
Dear Reviewer,

We thank the reviewers and the Editor for providing constructive and valuable comments and appreciate the opportunity to submit a revised version of our manuscript with the title 'SCHWANN CELL PLASTICITY REGULATES NEUROBLASTIC TUMOR CELL DIFFERENTIATION VIA EGFL8' by T. Weiss and S. Taschner-Mandl et al. for publication in Nature Communications.

Addressing the reviewers' comments, we have substantially revised the manuscript, clarified terminology, included quantifications, and now present additional new data that corroborate our initial findings.

We now include:

- 1. Data to support the axonal guidance properties of repair-related Schwann cells in neuroblastoma co-cultures and the novel, neuritogenic function of EGFL8 (Fig. 3 and 6).*
- 2. Additional IF stainings for key Schwann cell and repair Schwann cell regulators (JUN, SOX10), neuroblastoma markers (GD2) and T-cells (CD3) to better characterize cultured cells and tissues and provide thorough image quantification (Fig. 1,2, S.Fig. 1, 2, 5, 6).*
- 3. Control co-cultures with fibroblasts and neuroblastoma cells supporting that the pro-differentiating and anti-proliferative effects on neuroblastoma cells are specific to repair-related Schwann cells (S.Fig. 7).*
- 4. Proof for EGFL8 protein expression by stromal Schwann cells and repair Schwann cells (Fig. 7) and subcellular, vesicular localization (S.Video 1).*
- 5. First insights into the EGFL8 molecular mode-of-action by providing a map of the activated kinome in a time-resolved manner (Fig. 7, S.Fig. 13-15, S.Tables 7-10).*

A detailed point-by-point response is provided below. We believe that with the revised manuscript we went a significant step towards unraveling the down-stream signaling of EGFL8, a protein with up until now unassigned function.

REVIEWER COMMENTS

Reviewer #1 (Remarks to the Author):

In the manuscript entitled "Schwann cell plasticity regulates neuroblastic tumor cell differentiation via epidermal growth factor- like protein 8" by Weiss, Taschner Mandl et al. present a transcriptomics-driven study of the beneficial role of Schwann cells in benign tumors of the peripheral nervous system, leading to insight into the potential of Schwann-cell derived soluble factors to prevent or treat malignancies.

Starting from the notion that Schwann cells of supposedly non-tumor origin found in benign ganglioneuromas and absent from usually malign neuroblastomas might be a causative factor to the fate of these tumors, the authors performed RNA sequencing of patient samples of the tumor types along with injured nerves, isolated primary Schwann cells and neuroblastoma cell lines. A comparison of the transcriptomes revealed a repair-associated signature in ganglioneuromas resembling that of injured nerves and primary Schwann cells. As expected, as they are devoid of Schwann cells, the neuroblastoma samples clustered farther away from the Schwann-cell containing samples. To then

validate their concept of the role of Schwann cells in benign tumors and to evaluate a putative beneficial effect of Schwann cells on malignant tumors, the authors performed co-culture of primary Schwann cells with neuroblastoma cell lines. Indeed, the co-cultured neurons showed increased differentiation markers as well as a decrease in proliferation and an increase in apoptosis.

Importantly, the potentially beneficial effects on neuronal tumor cells were also present without direct contact of neurons and co-cultured Schwann cells. The authors therefore extracted potentially secreted factors from the RNAseq data and chose to test one of these factors, epidermal growth factor-like protein 8 (EGFL8), for effects on neuroblastoma cell lines. Treatment with recombinant EGFL8 resulted in concentration dependent pro-differentiating and anti-proliferating effects that were comparable with those of nerve growth factor. In line with a role of EGFL8 as a Schwann-cell secreted anti-tumor agent, EGFL8 protein was detected in supernatants of Schwann cell cultures, and high EGFL8 protein levels were measured in patient ganglioneuromas but not in neuroblastomas. This is a compelling study that is well performed. Most of the conclusions are soundly supported by data. The RNAseq data was adequately analysed.

The discovery of the roles of (repair) Schwann cells and secreted EGFL8 for tumor development is of utmost importance for both the cancer field and the glial physiology field. I find the manuscript clearly and well written, and I recommend it for publication after minor changes.

Author's response:

Dear Reviewer 1, we thank you for the thorough and thoughtful examination of the manuscript as demonstrated by your excellent summary of our work. Below, please find our responses to your comments.

Minor comments:

- The authors stress a **novel, neuritogenic function** of EGFL8 in several instances of the manuscript. However, I find the evidence presented not sufficient to support this claim. I see that neuroblastoma cells orient themselves and also project neurites along co-cultured Schwann cells (Fig. 3). Whether this effect is brought about in co-culture without direct contact or via EGFL8 treatment is not evident from the data. Fig. 6 does indicate an increase in amount and length of neurites after EGFL8 treatment, however these parameters would have to be quantified in order to provide evidence for EGFL8 as a neuritogen. The authors should either do this or refrain from this conclusion, which is not essential for the overall significance of the paper.

Author's response:

To support that human primary Schwann cells induce neurite outgrowth and alignment of neuroblastoma cells in co-cultures, we now have quantified the length as well as the orientation of neuritic processes of two neuroblastoma cell lines in co-cultures compared to control cultures without Schwann cells. The results demonstrate that both the length and alignment of the processes were significantly increased after co-culture with primary human Schwann cells (Fig. 3).

To explore whether the effect can be recapitulated by putative secreted Schwann cell derived factors, we showed in Fig 6 that treatment with EGFL8 induced neuronal differentiation (measured by NF200 expression) and reduced proliferation (measured by EdU incorporation) similar to NGF as determined by flow cytometry. To further support the claim that EGFL8 acts neuritogenic on neuroblastoma cells, we now have quantified the length of neuritic processes after EGFL8 treatment and compared it to neuroblastoma cells exposed to NGF (positive control) and without any treatment (negative control)

using the NeuronJ plugin of ImageJ. The results confirmed a significant increase in neurite length in response to NGF and EGFL8 treatment.

- There are a number of typos to correct, the authors should use a possible revision to thoroughly proof-read the text. Examples are found on p.6,l.18 (neuritogenic instead of neuritogentic) and l.25 (ganglioneuroblastomas instead of ganlgioneuroblastomas).

Author's response:

We extensively proof-read the text and corrected the typos.

Reviewer #2 (Remarks to the Author):

In this paper Weiss et al. explore the role of stromal cells in ganglioneuroma and neuroblastoma. They propose and begin to validate the interesting hypothesis that stromal cells resemble repair Schwann cells, and they promote more benign evolution of neural tumors. By comparing transcriptomic profile analysis of neuroblastoma (which have few stromal cells), ganglioneuroma (rich in stromal cells) and repair Schwann cells in injured nerves, the authors identify a common signature of stromal cells in ganglioneuroma and repair Schwann cells in injured nerve. They next analyze the secretome of repair Schwann cells, and among various factors they show that EGFL8 promote differentiation and decrease proliferation of a neuroblastoma tumor cell line. Interestingly, EGFL8 may also expressed by stromal cells. Overall the data are interesting and novel, and mostly convincing. This study uses human tissue and human cell lines and has direct clinical relevance. Moreover, the authors show an impressive correlation of tumor EGFL8 expression with patient survival rate. However the paper also suffers from some experimental weaknesses, that include lack of crucial quantifications and of some controls. These should be addressed to make the results stronger:

Author's response:

Dear Reviewer 2, we highly appreciate your effort and valuable comments to improve our manuscript! We edited the manuscript according to your suggestions and added substantial new data that corroborate our initial findings.

- the transcriptomic analysis is interesting, but given the complexity of the cellular composition of the different starting tissues, the data may be more difficult to interpret than the authors describe. Isolation of stromal cells and analysis of their transcriptome, or validation with single cell transcriptomic would strengthen the results. At minimum, confirmation of expression of EGFL8 by stromal cells (isolated, or in immunohistochemistry) would be essential.

Author's response:

We agree with the reviewer that the cellular composition of analyzed tissues is not discriminated using bulk RNA-sequencing analysis. However, we carefully characterized the used tissues for the Schwann cell content before the analysis (immunofluorescence staining for S100B) as demonstrated by the expression of specific neuroblastoma-associated genes (MYCN and LIN28B) and Schwann cell-associated genes (SOX10 and S100B) in the respective tissues. We now also provide immunofluorescence stainings for S100B and SOX10 on tissue sections to validate the Schwann cells with an additional marker and support the transcriptomic data (Fig. 1).

In addition, we included immunostainings of tissue sections that demonstrate the expression of EGFL8 in SOX10 positive stromal Schwann cells and repair Schwann cells on a cellular level (Fig. 7b,c).

In order to address your suggestion regarding single cell transcriptomics, we contacted the group of Dr. Ninib Baryawno who has recently published a pre-print (available under:

<https://www.biorxiv.org/content/10.1101/2020.05.04.077057v1>) that includes single-cell RNA-sequencing data on Schwann cell stroma-rich and poor human neuroblastic tumors. The authors kindly agreed to confidentially provide supporting data (see below, provided at the reviewer's discretion only). [REDACTED]

[IMAGE REDACTED]

- Figure 1 A-C requires quantification.

Author's response:

We have quantified the number of S100B⁺ Schwann cells and S100B⁻ cells on injured nerve sections as well as S100B⁺ Schwann cells, S100B⁻ cells, and NF200⁺ tumor cells in ganglioneuroma and neuroblastoma tumor sections and included the data in Supplementary Figure 1a:

- Figure 3. The alignment and increase neurite outgrowth should be quantified.

Author's response:

We now have quantified the length as well as the alignment of neuritic processes, which both were significantly increased after co-culture with primary human Schwann cells in Fig. 3.

- Can stromal cells be isolated and cultured, to show that they have the same effect of human Schwann cells, for example on the neuroblastoma alignment shown in figure 3?

Author's response:

We agree with the reviewer that culturing neuroblastoma cell lines with stromal Schwann cells derived from ganglioneuromas would be an informative experiment. Unfortunately, ganglioneuromas are a rare tumor entity and culturing the residing stromal Schwann cells in sufficient numbers required for the co-culture experiments was not possible.

- absence of GD2 expression in Schwann cells in figure 3F should be confirmed and shown.

Author's response:

We stained human Schwann cell cultures against GD2 and S100B and found that GD2 positive signals were occasionally found in Schwann cells (see Figure below, panel a, arrowheads). These GD2 signals could be derived from axonal debris that has been taken up by Schwann cells in culture. When Schwann cells were co-cultured with neuroblastoma cells, they apparently contained much more GD2 signals (see Figure below, panel b, arrowheads) - but this finding needs further validation and quantification. As this is not the main topic of the present study and we obtained new interesting data about the uptake of GD2 by Schwann cells, which is promising to result in a further manuscript, we now excluded the data from the manuscript. Importantly, this does not affect the interpretation of our FACS and immunofluorescence assays, since we focus our analysis on differentiation, apoptosis and proliferation of neuroblastoma cells, which are unequivocally identified as GD2 positive and S100B negative cells.

- Previous papers addressed the role of Schwann cells in neuroblastoma (e.g. Cross-talk between Schwann cells and neuroblasts influences the biology of neuroblastoma xenografts. Liu S, Tian Y, Chlenski A, Yang Q, Zage P, Salwen HR, Crawford SE, Cohn SL. *Am J Pathol.* 2005 Mar;166(3):891-900). These papers should be cited and detract a bit from the novelty.

Author's response:

Thank you for raising this point. We are aware of the study by Liu et al, as it demonstrated that neuroblastoma cells after being injected into peripheral nerve fascicles of mice formed less aggressive tumors that were infiltrated by Schwann cells. We now included the reference that must have been accidentally missed in the initial submission. Their finding supported the results from earlier studies published by us and others (cited in the introduction), which showed that the aggressiveness of neuroblastoma cell lines can be reduced upon exposure to Schwann cells and their secreted factors. However, the transcriptional program of Schwann cell stroma giving clues about whether they could have developed from adult Schwann cells that underwent repair-associated reprogramming has not been examined so far. The novelty of our study is demonstrated 1) by the evidence for a repair-associated phenotype of stromal Schwann cells as indicated by omics analyses of human tumor specimen, rich and poor in Schwann cell stroma and human injured nerves, rich in repair Schwann cells, and 2) by the identification and validation of a novel neuritogenic factor, EGFL8, found in stromal Schwann cells and repair Schwann cells, and its down-stream signaling, which likely promotes neuronal differentiation in both patho-physiological conditions.

- Figures 3, 4, 5, and throughout the manuscript: the authors frequently refer to the primary Schwann cells as repair Schwann cells. While primary Schwann cells share many features with repair Schwann cells, it is possible that these cells do not truly remain in the repair Schwann cell state once passaged in culture. For clarity, the authors could use the term primary Schwann cells to refer to these cells throughout the manuscript.

Author's response:

In a previous study we comprehensively characterized human Schwann cells in peripheral nerve explants and in primary cultures. Our transcriptomic and proteomic analyses showed that passage 1 Schwann cells exhibit a strikingly similar expression profile to repair Schwann cells residing in excised

nerve explants and also conduct repair-specific functions in vitro (1). Thus, we concluded that the Schwann cell repair program, once activated after axon loss, is dominating the Schwann cell state and persists in cultivated primary human Schwann cells, at least in passage one, that we used for the co-culture experiments. However, we agree with the reviewer that every in vitro culture will to a certain extent differ from the in vivo situation and we cannot be certain that all repair functions are preserved in primary human Schwann cell cultures. Therefore, we now introduce the term 'primary repair-related Schwann cells' when referring to cultured Schwann cells.

(1) Weiss, T., *et al.*, *Proteomics and transcriptomics of peripheral nerve tissue and cells unravel new aspects of the human Schwann cell repair phenotype. Glia*, 2016. **64**(12): p. 2133-2153.

- Figure 4B, the number of neurofilament positive cells should be quantified

Author's response:

For quantification of the flow cytometry data measuring NF200 expression in neuroblastoma cells, we used the NF200 mean fluorescence intensity and detected a shift from low/intermediate to high expression upon co-cultivation, which was confirmed by immunostainings (Fig 4.e-g). Therefore, we consider it not appropriate to draw an intensity level cut-off in order to calculate a percentage of positive cells. We now show the overlay with the histograms of unstained controls in addition to CTRL and co-cultured neuroblastoma cells in Fig 4c (CLB-Ma) and 4d (STA-NB-6) for better comparison.

- Jun is one of the key markers of repair Schwann cells. Immunohistochemistry showing JUN staining in SC-GN tissue would further validate the authors' transcriptomic findings.

Author's response:

We now provide immunostainings showing that SOX10 positive repair Schwann cells in injured peripheral nerves and stromal Schwann cells in ganglioneuromas are also positive for JUN in Fig 2d-f and Supplementary Figure 5:

- Olig1 and Shh are also key markers used to distinguish repair Schwann cells from other Schwann cell subtypes (Jessen and Mirsky, 2019, author's reference number 63). Did the authors examine these markers in their transcriptomic profiling of repair Schwann cells and stromal Schwann cells?

Author's response:

Indeed, SHH was significantly upregulated in the transcriptomic profile of stromal Schwann cells and repair Schwann cells and we now included the diagram in Fig 3c. Olig1 was unfortunately not annotated in the transcriptomic data.

- Fig. S1: the number of T cells present in neuroblastoma or human injured nerves should be checked.

Author's response:

We checked the number of T-cells present in neuroblastomas using immunostainings now shown in Supplementary Figure 4 (see below) and describe this in the results section. In line with the transcriptomic data, only few T-cells were present in neuroblastomas compared to ganglioneuromas. As injured nerves represent explant cultures that have been deprived of blood supply, we feel that determining T-cell numbers is not expressive.

- Is EGFL8 necessary and sufficient? would silencing of EGFL8 in stromal or Schwann cells diminish the effect of neuroblastoma cells?

Author's response:

We agree on the importance of mechanistic studies to address the relevance of EGFL8 in relation to other neuritogenic factors produced by Schwann cells. In our study, we stimulate neuroblastoma cell lines with recombinant EGFL8 and show, that EGFL8 is sufficient to induce neuronal differentiation (Fig 6a). In addition, we present new data in the revised manuscript on the down-stream signaling of EGFL8 in neuroblastoma cells. As no data currently exist on EGFL8 receptor or signaling, we employed an unbiased phospho-proteomics approach and now demonstrate that EGFL8 addition leads to specific phosphorylation of p38-MAPK-, MAPK7- and HIPK1-substrates only in the sensitive cell line STA-NB-6, but not in the insensitive STA-NB-10 (Fig 7h, S. Fig. 13-15, S.Tables 7-10). Interestingly, the re-wiring of cellular signaling by EGFL8 converged at known regulators of neurogenesis, such as PML (doi: 10.1158/0008-5472.can-03-1199) and NDRG2 (doi: 10.1016/j.neulet.2005.06.055.) (S. Fig. 13c, S.Tables 7-10), corroborating the role of EGFL8 as neuritogenic factor.

We also showed 1) that primary cultured as well as stromal Schwann cells upregulated a number of neuritogenic factors in addition to EGFL8, including NGF (Fig 2b), and 2) that a number of these, when added as recombinant factors, led to up-regulation of NF200 as measure for neuronal differentiation (Fig. 6b,c,e). Therefore, we assume that EGFL8 is a potent, but one of several factors mediating repair-related functions of Schwann cells.

Unfortunately, and despite our sincere efforts, in the present situation we were not able to obtain sufficient fresh nerve tissue to establish a robust protocol for genetic knock-down experiments in primary human Schwann cells and for the biological replicates necessary for co-cultivation. It will be

interesting to explore the question whether EGFL8 is necessary by silencing EGFL8 in human primary Schwann cells in future studies. We added a paragraph in the discussion section addressing this topic.

Minor points:

Figure 1 – For completeness and to demonstrate similarity of neuroblastoma cell lines to patient neuroblastoma tissue, could the authors also show immunostains of GD2 in ganglioneuroma and neuroblastoma tissue?

Author's response:

We now provide immunostainings showing that GD2 is highly present in neuroblastoma tumor tissue whereas it is only present on some ganglionic-like tumor cells in ganglioneuroma tissue in Supplementary Figure 1 b & c (see below).

The ability of repair SCs to take up GD2 (figure 3f) is an interesting finding. Could the authors comment further on this and whether (speculatively) stromal Schwann cells might act in a phagocytic manner in GN or NB tissue?

Author's response:

The ability of primary repair-related Schwann cells to take up the disialoganglioside GD2 indicates that their capacity of lipid uptake and degradation is not necessarily restricted to myelin. NB cells are known to abundantly express and release GD2 in form of micelles, monomers or membrane vesicles (1). The shedded tumor gangliosides were shown to bind and affect a variety of proteins and signaling molecules, to act immunosuppressive and help to escape immune recognition (2,3). If stromal Schwann cells indeed share the capacity of lipid uptake and degradation with repair Schwann cells, the resulting elimination of shedded GD2 may significantly impair the effect of neuroblastoma cells on the environment and attenuate tumor progression. As this topic is not within the main scope of this manuscript, we decided to rather not discuss the aspect of phagocytosis in this manuscript, but are currently about to prepare a manuscript further exploring this.

- 1) Ladisch, S., *et al.*, Shedding of GD2 ganglioside by human neuroblastoma. *Int J Cancer*, 1987. 39(1): p. 73-6.
- 2) Birkle, S., *et al.*, Role of tumor-associated gangliosides in cancer progression. *Biochimie*, 2003. 85(3-4): p. 455-63.

- 3) Lopez, P.H. and R.L. Schnaar, *Gangliosides in cell recognition and membrane protein regulation. Curr Opin Struct Biol*, 2009. 19(5): p. 549-57.

It is interesting that the transwell (i.e. indirect contact) seems to have a greater effect on NB differentiation in CLB-Ma and IMRS cell lines (figure 4b), yet direct contact has a greater effect on proliferation in these cell lines (figure 5b). Could the authors comment on whether they think this might be biologically relevant?

Author's response:

Thank you for commenting on this observation. We can only speculate that perhaps direct cell-cell contact provides additional signaling halting cell cycle independent of differentiation and at the same time dampening differentiation related signals, that are in turn effective without direct contact in these two cell lines. Postulating cross-talk between Schwann and neuroblastoma cells, we can also speculate that direct contact may – depending on the cell line - alter the Schwann cell expression profile in a different way as compared to transwell cultures. In turn, this could reflect back on differences in neuroblastoma proliferation vs differentiation in direct and indirect cultures.

In addition, the five neuroblastoma cell lines used for this study do not only differ in MYCN amplification status, but in several additional genetic and potentially epi-genetic alterations which will affect gene expression and thus the entire signaling network controlling differentiation and proliferation. A much larger number of cell lines with similar genetic makeup as CLB-Ma and IMR5 would be needed to draw biologically sound conclusions. We would thus prefer to not speculate on these observations in the manuscript.

The role and expression of HLA in Schwann cells has been extensively documented by many papers, some of them just from a quick search are listed below. HLA expression by Schwann cells is not so novel and some of these papers should be referenced.

Author's response:

We agree with the reviewer that many studies have shown that Schwann cells are able to express MHCII proteins and now included appropriate references in the revised manuscript. However, upregulation of MHCII on Schwann cells was primarily reported in (auto-) inflammatory or infectious neuropathies and upon treatment with INF γ . The immunoregulatory role of MHCII expressing Schwann cells during the regeneration of injured nerves or in ganglioneuromas remains to be evaluated.

Page 3, third bullet point – “capable to induce neuronal differentiation” reads better as “capable of inducing neuronal differentiation”

Author's response:

We thank the reviewer for careful reading and changed the phrase accordingly.

Page 4, abbreviations section – typo “ganglioneuroblatoma” should be “ganglioneuroblastoma”

Author's response:

Thank you, we corrected the typo.

Page 6 – typo “ganlioneuroblastomas” should be “ganglioneuroblastomas”

Author's response:

Thank you, we corrected the typo

Page 9 – in the text, could the authors clarify that the active protein modification and transport machinery is in repair SCs, rather than stromal SCs?

Author's response:

We revised the sentence to clarify that statement.

Page 10 – “alongside (with) controls for 11 days”; could the authors state clearly in the text what the controls were (i.e. Schwann cells or NBT cells alone). Also this phrase might read more easily as “alongside controls for 11 days”.

Author's response:

We adapted the sentence accordingly.

Page 13/ figure 6g – could the authors be clear as to why the human data sets comprise 498 (presumably corresponding to supp fig 4?) and 649 tumor specimens, but only 471 patient survival outcomes are reported in 6g?

Author's response:

We presented 2 transcriptomic datasets of neuroblastic tumors, 1) the Kocak dataset, containing 649 tumor specimens (Fig. 6h, S.Fig. 10a), and 2) the SEQC dataset, containing 498 tumor specimens (S.Fig. 10b) in the manuscript. Those tumors are derived from biopsies or surgeries but the survival data are not available for all patients. We now included the Kaplan-Meier survival curves and expression graph of both datasets in the Supplementary figure 10. The information on the number of patients with survival data is indicated by the shown n number. Note that we also corrected the number of patients with high EGFL8 expression from 250 to 255 in Fig. 6h.

Page 14 – authors switch from using “vimentin” to “vime” but should be consistent throughout the manuscript

Author's response:

We now consistently use ‘VIME’.

Figure 1 – typo in 1b bahavior should be behavior

Author's response:

Thank you, we corrected the typo.

Figure 1 legend – please could the authors define the acronym MNA

Author's response:

The acronym MNA means ‘MYCN amplified’ and is now defined in the figure legend.

Figure 1 legend – the n numbers reported in the legend for RNA seq do not appear to match up with the data represented in the figure

Author's response:

Please note that the NB cell line STA-NB-6 was analyzed in three biological replicates and that one Schwann cell culture was measured in two biological replicates, so the stated n number is correct. This is now indicated in the figure legend.

Figure 3 – could the authors include a low magnification panel of SC/CLB-Ma co-cultures, similar to in 3d?

Author's response:

We restructured the Figure and now show low and high magnification images of Schwann cell/STA-NB-6 and Schwann cell/CLB-Ma co-cultures.

Figure 3 legend – typo (b) GD2+/vimentin~ should be (b) GD2+/vimentin+

Author's response:

As the vimentin channel was removed for better visualization of the neurites, also the typo was removed.

Figure 7 – could the authors include a video of confocal Z stacks to further illustrate the subcellular localization of EGFL8 vesicular structures?

Author's response:

We performed new immunofluorescence stainings for Figure 7, which now shows Schwann cells stained for Schwann cells membrane marker NGFR and EGFL8 and include a video of the Z stacks to visualize the subcellular location of EGFL8 signals.

Figure 7D – could the authors add arrows to the blots to indicate the EGFL8 bands?

Author's response:

We now indicate the expected EGFL8 bands at 32 kDa (filled white arrowheads) and the additional band at about 37 kDa (lined white arrowheads) that could represent the EGFL8 protein with posttranslational modifications.

Figure 7D – typo third lane heading of first blot and all headings of second blot: “culure” should be “culture”

Author's response:

Thank you, we corrected the typo.

Figure 7D – while I appreciate that this may have been a difficult antibody to work with, the bands on the Western blots are not crisp; do the authors have better example blots they could use in this figure?

Author's response:

Indeed, EGFL8 is a yet poorly characterized protein. We included the whole western blots (Ponceau staining, immunofluorescence detection of GAPDH, chemiluminescence detection of EGFL8) in the Supplementary Figure 11 (see below) for detailed examination; filled white arrowheads indicate bands of about 32 kDa, the proposed molecular weight of EGFL8, lined white arrowheads indicate bands of about 37 kDa that could represent the EGFL8 protein with posttranscriptional modifications. As the GAPDH control bands from the same blots are crisp, and the antibody recognized the recombinant EGFL8 protein control, we assume that the achieved EGFL8 bands are representative.

Western Blot analysis of EGFL8 protein expression

Materials and methods pg 20 – define FBs

Author’s response:

The term ‘FBs’ is now defined as fibroblasts.

Materials and methods pg 21 – supplementary tables 1 and 2 show NB and GN tumor characteristics, supplementary table 3 is antibodies

Author’s response:

Thank you, we corrected this.

HLA in Schwann cell references:

Expression of antigen processing and presenting molecules by Schwann cells in inflammatory neuropathies. Meyer Zu Horste G, Heidenreich H, Lehmann HC, Ferrone S, Hartung HP, Wiendl H, Kieseier BC. *Glia*. 2010 Jan 1;58(1):80-92. doi: 10.1002/glia.20903.

Immunohistochemical localizations of class II antigens and nerve fibers in human carious teeth: HLA-DR immunoreactivity in Schwann cells. Yoshida N, Yoshida K, Iwaku M, Ozawa H. *Arch Histol Cytol*. 1998 Oct;61(4):343-52.

Axonal neuropathy associated with interferon-alpha treatment for hepatitis C: HLA-DR immunoreactivity in Schwann cells. Quattrini A, Comi G, Nemni R, Martinelli V, Villa A, Caimi M, Wrabetz L, Canal N. *Acta Neuropathol.* 1997 Nov;94(5):504-8.

LA-DR expression in peripheral neuropathies: the role of Schwann cells, resident and hematogenous macrophages, and endoneurial fibroblasts. Sommer C, Schröder JM. *Acta Neuropathol.* 1995;89(1):63-71

Class II antigen expression in peripheral neuropathies. Mitchell GW, Williams GS, Bosch EP, Hart MN. *J Neurol Sci.* 1991 Apr;102(2):170-6.

Class II antigen expression on human cultured Schwann cells from patients with Charcot-Marie-Tooth disease. De Martini I, Bianchini D, Schenone A, Cadoni A, Zicca A, Zaccheo D, Mancardi GL. *Neurosci Lett.* 1989 May 22;100(1-3):331-4.

HLA-DR Schwann cell reactivity in peripheral neuropathies of different origins. Mancardi GL, Cadoni A, Zicca A, Schenone A, Tabaton M, De Martini I, Zaccheo D. *Neurology.* 1988 Jun;38(6):848-51.

HLA-DR-expressing cells and T-lymphocytes in sural nerve biopsies. Schröder HD, Olsson T, Solders G, Kristensson K, Link H. *Muscle Nerve.* 1988 Aug;11(8):864-70.

Reviewer #3 (Remarks to the Author):

The study of Weiss and colleagues starts from the observation that ganglioneuromas, which contain a Schwann cell-rich stroma, are more benign than related neuroblastomas, which have very low numbers of Schwann cells. To study Schwann cell properties and their role in these tumors, the authors generated expression profiles, compared them with expression profiles of injured nerves and concluded that stromal Schwann cells are very similar to repair Schwann cells in the nerve. They went on to show that primary Schwann cells stimulate differentiation, inhibit proliferation and induce cell death in several neuroblastoma cell lines, and that Schwann-cell produced and secreted EGFL8 is responsible for at least part of the pro-differentiating effect. This leads them to conclude that the repair phenotype allows Schwann cells to influence tumor cell behavior.

The study is interesting in principle and yields insights into the modulatory role of Schwann cells in neural crest-derived tumors. The manuscript is professionally put together and well written. Data are clearly presented. Still I find several issues problematic.

Author's response:

We thank the reviewer for assessing our manuscript as interesting and well written and appreciate critical comments and valuable suggestions. In the revised manuscript we have addressed all comments as detailed below.

In their RNA-Seq studies, the authors generate expression profiles of tumors and injured nerve, then subtract neuroblastoma-expressed genes from those expressed in ganglioneuromas or injured nerve, and compared the two gene sets. Considering that the biggest difference between ganglioneuromas and neuroblastomas are Schwann cells that also make up a large portion of the cells in the injured nerve, it is almost a self-fulfilling prophecy that the authors detect a common Schwann cell-specific signature in both analyzed groups. Whether this common signature is that of a repair Schwann cell is, however, not fully substantiated. From the absence of myelin in the tumors, it is obvious that the Schwann cells within the tumor are not myelinating Schwann cells, but whether they are really repair Schwann cells or some other Schwann cell type (such as immature Schwann cells that are present during ontogenetic development) is difficult to judge. In support of their assumption, the authors point out that the signature of stromal Schwann cells resembles that of primary cultured Schwann cells, which they claim to closely resemble repair Schwann cells. They even go as far as referring to cultured primary Schwann cells as cultured repair Schwann cells. To my mind this is not justified. It is obvious that Schwann cells in cell culture have a number of properties that they share with repair Schwann cells. However, they share these same properties with all other types of proliferative Schwann cells. This may all seem like semantics. However, these semantics are important for this paper, because the fundamental claim of the paper builds on it.

Author's response:

Indeed, repair Schwann cells re-express several markers that are also present in developing Schwann cells (precursors/immature subtypes). However, it has been demonstrated that repair Schwann cells also express repair-specific genes such as GDNF and SHH and acquire novel, repair-specific functions that distinguish proliferating repair Schwann cells after injury from proliferating Schwann cells during development (1,2,3,4). Those functions include myelin degradation, the secretion of chemokines and cytokines for immune cell attraction, the formation of regeneration tracks for axon guidance, MHCII upregulation, and expression of distinct neurotrophic factors (3,5). We revised the introduction and elaborate on the difference between repair Schwann cells to developing Schwann cells in more detail.

Furthermore, novel single-cell RNA-sequencing data derived from neonatal, uninjured (adult) and injured mouse peripheral nerves have been recently published that suggest further genes specifically expressed by Schwann cells upon injury (4). Importantly, we found the majority of them significantly expressed in the enriched gene set of repair Schwann cells and stromal Schwann cells (Fig. 2c) and now present these findings in addition to the indicated repair-specific functions determined by the functional annotation analysis (Fig. 2b).

Although similarities between the enriched gene sets of repair Schwann cells and stromal Schwann cells are expected due to the prevalent Schwann cell content, a repair-associated cell state of stromal Schwann cells is now supported by the presence of genes that characterize the repair Schwann cell state involving 1) genes associated with developing Schwann cells (S.Fig. 4a,b), and 2) genes specific for repair Schwann cells (Fig. 2c). In addition, we validated JUN, the key transcription factor determining the repair SC identity in stromal SCs (Fig. 2d,e).

In a previous study we comprehensively characterized human Schwann cells in peripheral nerve explants and in primary cultures. Our transcriptomic and proteomic analyses showed that passage 1 Schwann cells exhibit a strikingly similar expression profile to repair Schwann cells residing in excised nerve explants and also conduct repair-specific functions such as myelin clearance in vitro (5). Thus, we concluded that the Schwann cell repair program, once activated after axon loss, is dominating the Schwann cell state and persists in primary human Schwann cell cultures, at least in passage one, that we used for our co-culture experiments. However, we agree with the reviewer that every in vitro culture will to a certain extent differ from the in vivo situation and we cannot be certain that all repair functions are preserved in primary human Schwann cells. Therefore, we now use the term 'primary repair-related Schwann cells' when referring to cultured primary Schwann cells and we now elaborate in more detail on what discriminates human repair and other Schwann cell types in the introduction and discussion.

The presence of repair-like SCs in GNs also implicates that the tumor cells express factors able to induce and maintain a repair-like SC state in the microenvironment. Thus, we propose that stromal SCs originate from adult SCs that react to peripheral neuroblastic tumor cells in a similar way as to injured neurons. Moreover, the progressing death of ganglionic-like tumor cells and resulting axon degeneration observed in GNBs/GNs (6) could supply stromal SCs with cues that trigger the repair-like state and explain why it does not diminish over time. As a consequence, stromal SCs could continuously exert nerve repair-associated functions in the microenvironment that are responsible for a benign tumor development.

- (1) Arthur-Farraj, P.J., et al., c-Jun reprograms Schwann cells of injured nerves to generate a repair cell essential for regeneration. *Neuron*, 2012. 75(4): p. 633-47.
- (2) Jessen, K.R., R. Mirsky, and P. Arthur-Farraj, The Role of Cell Plasticity in Tissue Repair: Adaptive Cellular Reprogramming. *Dev Cell*, 2015. 34(6): p. 613-20.
- (3) Jessen, K.R. and R. Mirsky, The repair Schwann cell and its function in regenerating nerves. *J Physiol*, 2016. 594(13): p. 3521-31.
- (4) Toma, J.S., et al., Peripheral Nerve Single-Cell Analysis Identifies Mesenchymal Ligands that Promote Axonal Growth. *eNeuro*, 2020. 7(3).
- (5) Weiss, T., et al., Proteomics and transcriptomics of peripheral nerve tissue and cells unravel new aspects of the human Schwann cell repair phenotype. *Glia*, 2016. 64(12): p. 2133-2153.
- (6) Li, Y. and A. Nakagawara, Apoptotic cell death in neuroblastoma. *Cells*, 2013. 2(2): p. 432-59.

For analyzing the effect of Schwann cells on tumor cells (Figs. 4,5; S.Fig. 2), the authors exclusively rely on neuroblastoma cell lines (whose response varies substantially). Long established neuroblastoma cell lines have adjusted to growth in culture and may substantially differ in their response from primary tumor cells. This has not been taken into consideration.

Author's response:

We agree to the general notion that cell lines cannot fully reflect the situation in the tumor. In our study we aimed to address the limitations of in vitro models at several levels:

- 1) in addition to three long established cell lines, SH-SY-5Y, CLB-Ma and IMR5 that are commonly used to investigate neuroblastoma cell behavior, we included patient-derived, low-passage neuroblastoma cultures (STA-NB-6 and STA-NB-10. Interestingly, we observed similar results in both, short-term neuroblastoma cultures and established neuroblastoma cell lines and demonstrated that the response was rather dependent on MYCN amplification status than on primary cultures vs established cell lines (see Fig. 4, 5, 6). We have also compared genetic copy number profiles of cultures and corresponding tumors confirming that short term cultures reflect the genomic profiles found in primary tumors (data can be provided upon request). We also now better describe the patient-derived cultures in the material and methods as well as in the results section.*
- 2) In our co-cultivation models we only use primary Schwann cells in passage 1 and not immortalized Schwann cells or cell lines, as in our experience these differ substantially in the expression of key Schwann cell markers.*

As carried out there is furthermore no evidence that the observed pro-differentiating, anti-proliferating and pro-apoptotic effects are specific to Schwann cells. How can the authors exclude that fibroblasts (or any other cell type) do not have the same effects when co-cultured with neuroblastoma cells?

Author's response:

We agree with the reviewer that analyzing the proliferation and neuronal differentiation status of neuroblastoma cell line and primary cultures after co-culture with other cell types will serve as valid control. Therefore, we repeated the experiment and co-cultured STA-NB-6 and SH-SY5Y cells (that showed the strongest response to Schwann cells) with immortalized human fibroblasts (iFBs) and primary cancer associated FBs (CAFs). After 16 days, the NF200 expression as measure of differentiation was either unaffected or even significantly decreased in both, STA-NB-6 and SH-SY-5Y, after direct and indirect co-culture with iFBs as well as CAFs. The co-cultures did not cause any significant changes in the proliferation rate of neuroblastoma cells. We can now exclude that fibroblasts have the same effect on neuroblastoma cells as Schwann cells. These results support that the observed pro-differentiating, anti-proliferative effects are specific to Schwann cells and were included in S.Fig. 7 (see below).

In addition, we have performed co-cultivation experiments with peripheral blood mononuclear cells from healthy donors within the scope of another study and did not observe changes in neuroblastoma cell proliferation (data can be provided upon request).

Among the Schwann cell effects on tumor cells, the pro-differentiating effect is probably the most specific (Fig. 4, see above). When looking at the data, it is also the least robust. Therefore, it would be reassuring to see the pro-differentiating effect confirmed in immunocytochemical stainings with several markers. In the current state, most immunocytochemical analyses throughout the paper are carried out with a very limited set of markers, mostly NF200 as neuronal marker and S100B as Schwann cell marker.

Author's response:

We cannot completely agree with the reviewer's comment on NF200, as this is an established marker to identify mature axons (1). NF200 expression was significantly elevated in neuroblastoma cells co-cultured with Schwann cells compared to neuroblastoma cells cultured alone as quantified by FACS (Fig. 4 b,c,d). Furthermore, we validated and illustrated the expression of NF200 in the long neurite projections of GD2 positive neuroblastoma cells shown in immuno-cytostainings upon co-culture with Schwann cells (Fig. 4e,g), while only a low expression of NF200 was evident in neuroblastoma cell controls (Fig. 4f,h). We now quantified the neurite length as additional marker for neuronal differentiation. The significant increase in neurite length (Fig. 3i) together with a significantly higher expression of NF200 (Fig. 4) in neuroblastoma cells upon co-culture with Schwann cells confirms the validity and robustness of our neuronal differentiation assay.

To provide further markers characteristic for Schwann cells, we now included immuno-cytostainings of primary Schwann cells using Schwann cell markers SOX10 (Fig. 3b, see first panel below) and NGFR (Fig. 7f, see second panel below) in addition to S100B.

- 1) Trojanowski, J.Q., N. Walkenstein, and V.M. Lee, Expression of neurofilament subunits in neurons of the central and peripheral nervous system: an immunohistochemical study with monoclonal antibodies. *J Neurosci*, 1986. 6(3): p. 650-60.

Mechanistic data are missing on the molecular mode of EGFL8 action.

Author's response:

To address this, we present new data in the revised manuscript on the down-stream signaling of EGFL8 in neuroblastoma cells. As no data currently exist on EGFL8 receptor or signaling in human or any other mammalian cells, we employed an unbiased phospho-proteomics approach in the EGFL8 responsive STA-NB-6 neuroblastoma versus non-responsive STA-NB-10 cells in a time-resolved manner. We demonstrate that EGFL8 addition leads to specific phosphorylation of p38 α /MAPK-, ERK5/MAPK7- and HIPK1-substrates only in the sensitive cell line STA-NB-6, but not in the insensitive STA-NB-10 (Fig 7h, S.Table 7-10 and S.Fig. 13). Interestingly, the re-wiring of cellular signaling by EGFL8 converged at known regulators of neurogenesis, such as PML (doi: 10.1158/0008-5472.can-03-1199) and NDRG2 (doi: 10.1016/j.neulet.2005.06.055.) (Fig 7i, S.Table 7-10 and S.Fig. 13), corroborating the role of EGFL8 as neuritogenic factor. The results and discussion section have been adapted accordingly.

Other issues:

Fig.1a and p.8: The authors state that degrading axons in injured nerves are identified by NF200 staining. NF200 labels nerves, not specifically degrading ones.

Author's response:

We agree with the reviewer that the sentence can be misunderstood. We rephrased the sentence to make clear that axons are identified by NF200 expression and that the axons of injured nerves have mostly disintegrated after the degeneration period of 7 days.

Fig.2: The authors argue that that the common signature obtained from comparing tumors and injured nerves is a Schwann cell signature. While this is true for the most part, some aspects of the signature are also contributed by immune cells that are present in injured nerves and differentially occur in ganglioneuromas vs. neuroblastomas. Therefore the authors cannot formally exclude that terms associated to immune regulation, phagocytosis etc. are enriched in the signature because of immune rather than Schwann cells. Further experiments are needed.

Author's response:

Indeed, the functional annotation terms found in the shared ganglioneuroma/injured nerve expression profile cannot be exclusively attributed to Schwann cells. To further validate the transcriptomic findings of high interest on a cellular level, we now provide immunofluorescence stainings that confirm the expression of HLA-DR, SOX10, and JUN in repair and stromal Schwann cells (Fig. 1i-l, Fig. 2d-g, S.Fig. 2,3,4).

Fig. 3d,e: These experiments additionally require a quantitative read-out.

Author's response:

We included the quantification of the length as well as neurite alignment of neuritic processes of neuroblastoma cells in co-cultures with Schwann cells compared to neuroblastoma cells cultured without Schwann cells. The results demonstrated a significant increase of neurite length and alignment in response to primary human Schwann cells now shown in Fig. 3i & j.

Fig. 7d: What is the 55 kDa band in some of the Schwann cell lysates. It is difficult to judge whether the approx. 35 kD band in lysates and supernatants is the same, because samples are on different gels with different running behavior.

Author's response:

We now provide the entire Western blot data (including Ponceau staining, immunofluorescence detection of GAPDH, chemiluminescence detection of EGFL8) in the Supplementary Figure 11 (see below) for detailed examination; filled white arrowheads indicate bands of about 32 kDa, the proposed molecular weight of EGFL8, lined white arrowheads indicate bands of about 37 kDa that might represent the EGFL8 protein with posttranscriptional modifications. We cannot provide an answer for the 55 kDa band in the SC lysates but speculate that this band represents EGFL8-protein complexes that have not been fully reduced and denatured. The different running behavior is explained by the high load of proteins in the culture supernatant samples.

Western Blot analysis of EGFL8 protein expression

There is no evidence for a role of EGFL8 on nerve regeneration. This would be an important piece of data as the authors imply similar function of the same cell type in injury and tumors. However, I understand that this may be beyond the scope of the study.

Author’s response:

We identified in a previous study that EGFL8 is significantly upregulated in injured nerves compared to uninjured controls (1). Moreover, EGFL8 shares similar domains and molecular weight with EGFL7 (2) which was described to induce neural stem cell differentiation (3). Hence, EGFL8 may act in a similar way to promote axon differentiation after peripheral nerve injury. In the revised manuscript, we now include these points.

- (1) Weiss, T., *et al.*, Proteomics and transcriptomics of peripheral nerve tissue and cells unravel new aspects of the human Schwann cell repair phenotype. *Glia*, 2016. **64**(12): p. 2133-2153.
- (2) Fitch MJ, Campagnolo L, Kuhnert F, Stuhlmann H. 2004. *Egfl7*, a novel epidermal growth factor-domain gene expressed in endothelial cells. *Dev Dyn* 230:316–324
- (3) Schmidt MH, Bicker F, Nikolic I, Meister J, Babuke T, Picuric S, Muller-Esterl W, Plate KH, Dikic I. 2009. Epidermal growth factor-like domain 7 (EGFL7) modulates Notch signalling and affects neural stem cell renewal. *Nat Cell Biol* 11:873–880

Reviewer #1 (Remarks to the Author):

The authors have addressed my concerns sufficiently. In the title of the y-axis of Fig. 3j, please replace “varianz” with “variance”.

Reviewer #2 (Remarks to the Author):

We think that the authors did a pretty good job at responding to a lot of the remarks, and for the ones that they cannot, they give fairly convincing reasoning.

We have a minor comment on one new figure:

Supplementary figure 1a: Could the authors label the Y axis for clarity that this is the percentage of cells as opposed to absolute number?

Reviewer #3 (Remarks to the Author):

The authors have satisfactorily addressed my comments. I now recommend acceptance and publication of the manuscript as is.

Reviewer #4 (Remarks to the Author):

Weiss et al. carefully addressed in their revised version of the manuscript “Schwann cell plasticity regulates neuroblastic tumor cell differentiation via epidermal growth factor like protein 8” the comments of the three initial reviewers. The manuscript has markedly improved and most critical points of the reviewers have been clearly solved.

The findings are of high relevance for the neuroblastoma research field, as they describe a strong impact of tumor-infiltrating Schwann cells and, in particular, EGFL8 secreted by Schwann cells in the biology of low-risk neuroblastoma which show more differentiated and biologically favorable clinical phenotype.

Still, I have one critical point which should be addressed before final acceptance. The authors describe two data sets analyzed via the R2 database, which they describe as two different (independent) studies (Figure 6 h, Figure Supplementary Figure 11): the Kocak and the SEQC cohorts. Both data sets are not independent and have a large overlap of neuroblastoma tumors, which have been simply analyzed by two different methods (Kocak=oligo array and SEQC=RNAseq). I would strongly recommend to show results only from one of these data sets. If the authors want to include results from a truly independent cohort, I would recommend to additionally analyze the NRC cohort (GEO85047).

If this critical point is fixed, I strongly recommend to accept this exciting paper for publication in Nature Communications.

Vienna, Dec 8th 2020

Dear Editor,
Dear Reviewers,

We thank the Reviewers and the Editor for providing positive feedback and valuable comments and appreciate the opportunity to submit a revised version of our manuscript with the title 'SCHWANN CELL PLASTICITY REGULATES NEUROBLASTIC TUMOR CELL DIFFERENTIATION VIA EGFL8' by T. Weiss and S. Taschner-Mandl *et al.* for publication in Nature Communications.

Addressing the reviewers' comments, we revised the manuscript. Please, find a detailed point-by-point response provided below.

REVIEWER COMMENTS

Reviewer #1 (Remarks to the Author):

The authors have addressed my concerns sufficiently. In the title of the y-axis of Fig. 3j, please replace "varianz" with "variance".

Response: We have corrected the y-axis label of Fig.3j.

Reviewer #2 (Remarks to the Author):

We think that the authors did a pretty good job at responding to a lot of the remarks, and for the ones that they cannot, they give fairly convincing reasoning.

We have a minor comment on one new figure:

Supplementary figure 1a: Could the authors label the Y axis for clarity that this is the percentage of cells as opposed to absolute number?

Response: We have corrected the y-axis label in Supplementary Figure 1a accordingly.

Reviewer #3 (Remarks to the Author):

The authors have satisfactorily addressed my comments. I now recommend acceptance and publication of the manuscript as is.

Reviewer #4 (Remarks to the Author):

Weiss *et al.* carefully addressed in their revised version of the manuscript "Schwann cell plasticity regulates neuroblastic tumor cell differentiation via epidermal growth factor

like protein 8” the comments of the three initial reviewers. The manuscript has markedly improved and most critical points of the reviewers have been clearly solved.

The findings are of high relevance for the neuroblastoma research field, as they describe a strong impact of tumor-infiltrating Schwann cells and, in particular, EGFL8 secreted by Schwann cells in the biology of low-risk neuroblastoma which show more differentiated and biologically favorable clinical phenotype.

Still, I have one critical point which should be addressed before final acceptance. The authors describe two data sets analyzed via the R2 database, which they describe as two different (independent) studies (Figure 6 h, Figure Supplementary Figure 11): the Kocak and the SEQC cohorts. Both data sets are not independent and have a large overlap of neuroblastoma tumors, which have been simply analyzed by two different methods (Kocak=oligo array and SEQC=RNAseq). I would strongly recommend to show results only from one of these data sets. If the authors want to include results from a truly independent cohort, I would recommend to additionally analyze the NRC cohort (GEO85047).

If this critical point is fixed, I strongly recommend to accept this exciting paper for publication in Nature Communications.

***Response:** Thank you very much for your valuable comment and suggestion! We have now removed the SEQC dataset and kept the Kocak dataset. In addition, as recommended by the reviewer, we now also include the NRC dataset and show the results in Supplementary Figure 11.*